# FlexEControl: Flexible and Efficient Multimodal Control for Text-to-Image Generation

## Abstract

Controllable text-to-image (T2I) diffusion models generate images conditioned on both text prompts and semantic inputs of other modalities like edge maps. Nevertheless, current controllable T2I methods commonly face challenges related to efficiency and faithfulness, especially when conditioning on multiple inputs from either the same or diverse modalities. In this paper, we propose a novel *Flexible* and *Efficient* method, FlexEControl, for controllable T2I generation. At the core of FlexEControl is a unique weight decomposition strategy, which allows for streamlined integration of various input types. This approach not only enhances the faithfulness of the generated image to the control, but also significantly reduces the computational overhead typically associated with multimodal conditioning. Our approach achieves a reduction of 41% in trainable parameters and 30% in memory usage compared with Uni-ControlNet. Moreover, it doubles data efficiency and can flexibly generate images under the guidance of multiple input conditions of various modalities.

## 1 Introduction

In the realm of text-to-image (T2I) generation, diffusion models exhibit exceptional performance in transforming textual descriptions into visually accurate images. Such models exhibit extraordinary potential across a plethora of applications, spanning from content creation (Rombach et al., 2022; Saharia et al., 2022b; Nichol et al., 2021; Ramesh et al., 2021a; Yu et al., 2022; Avrahami et al., 2023; Chang et al., 2023), image editing (Balaji et al., 2022; Kawar et al., 2023; Couairon et al., 2022; Zhang et al., 2023; Valevski et al., 2022; Nichol et al., 2021; Hertz et al., 2022; Brooks et al., 2023; Mokady et al., 2023), and also fashion design (Cao et al., 2023). We propose a new unified method that can tackle two problems in text-to-image generation: improve the training efficiency of T2I models concerning memory usage, computational requirements, and a thirst for extensive datasets (Saharia et al., 2022a; Rombach et al., 2022; Ramesh et al., 2021b); and improve their controllability especially when dealing with multimodal conditioning, e.g. multiple edge maps and at the same time follow the guidance of text prompts, as shown in Figure 1 (c).

Controllable text-to-image generation models (Mou et al., 2023) often come at a significant training computational cost, with linear growth in cost and size when training with different conditions. Our approach can improve the training efficiency of existing text-to-image diffusion models and unify and flexibly handle different structural input conditions all together. We take cues from the efficient parameterization strategies prevalent in the NLP domain (Pham et al., 2018; Hu et al., 2021; Zaken et al., 2021; Houlsby et al., 2019) and computer vision literature (He et al., 2022). The key idea is to learn shared decomposed weights for varied input conditions, ensuring their intrinsic characteristics are conserved. Our method has several benefits: It not only achieves greater compactness (Rombach et al., 2022), but also retains the full representation capacity to handle various input conditions of various modalities; Sharing weights across different conditions contributes to the data efficiency; The streamlined parameter space aids in mitigating overfitting to singular conditions, thereby reinforcing the flexible control aspect of our model.

Meanwhile, generating images from multiple homogeneous conditional inputs, especially when they present conflicting conditions or need to align with specific text prompts, is challenging. To further augment our model's capability to handle multiple inputs from either the same or diverse modalities as shown in Figure 1,

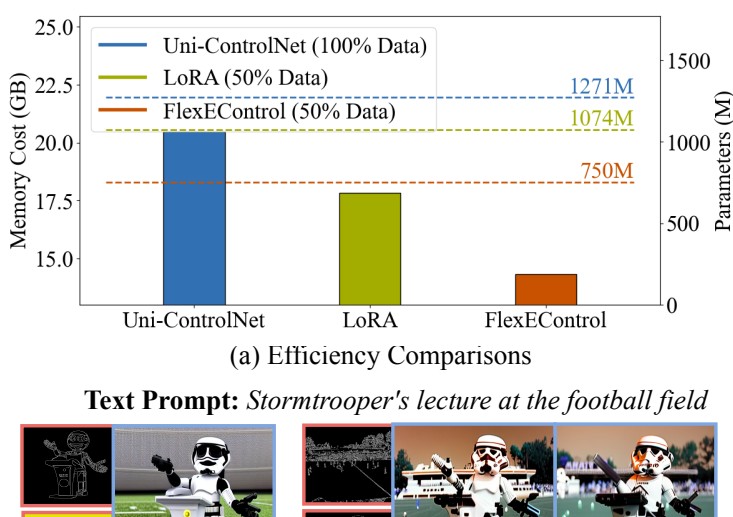

(a) Efficiency Comparisons

**Text Prompt:** *Stormtrooper's lecture at the football field*

(b) Controllable T2I w.
Different Input Conditions

(c) Controllable T2I w.
Same Input Conditions

Figure 1: (a) FlexEControl excels in training efficiency, achieving superior performance with just half the training data compared to its counterparts on (b) Controllable Text-to-Image Generation w. Different Input Conditions (one edge map and one segmentation map). (c) FlexEControl effectively conditions on two canny edge maps. The text prompt is `Stormtrooper's lecture at the football field` in both Figure (b) and Figure (c).

during training, we introduce a new training strategy with two new loss functions introduced to strengthen the guidance of corresponding conditions. This approach, combined with our compact parameter optimization space, empowers the model to learn and manage multiple controls efficiently, even within the same category (e.g., handling two distinct segmentation maps and two separate edge maps). Our primary contributions are summarized below:

- We propose FlexEControl, a novel text-to-image generation model for efficient controllable image generation that substantially reduces training memory overhead and model parameters through decomposition of weights shared across different conditions.

- We introduce a new training strategy to improve the flexible controllability of FlexEControl. Compared with previous works, FlexEControl can generate new images conditioning on multiple inputs from diverse compositions of multiple modalities.

- FlexEControl shows on-par performance with Uni-ControlNet (Zhao et al., 2023) on controllable text-to-image generation with 41% less trainable parameters and 30% less training memory. Furthermore, FlexEControl exhibits enhanced data efficiency, effectively doubling the performance achieved with only half amount of training data.

## 2 Method

The overview of our method is shown in Figure 2. In general, we use the copied Stable Diffusion encoder (Stable Diffusion encoder block and Stable Diffusion middle block) which accepts structural conditional input and then perform efficient training via parameter reduction using Kronecker Decomposition first (Zhang et al.,

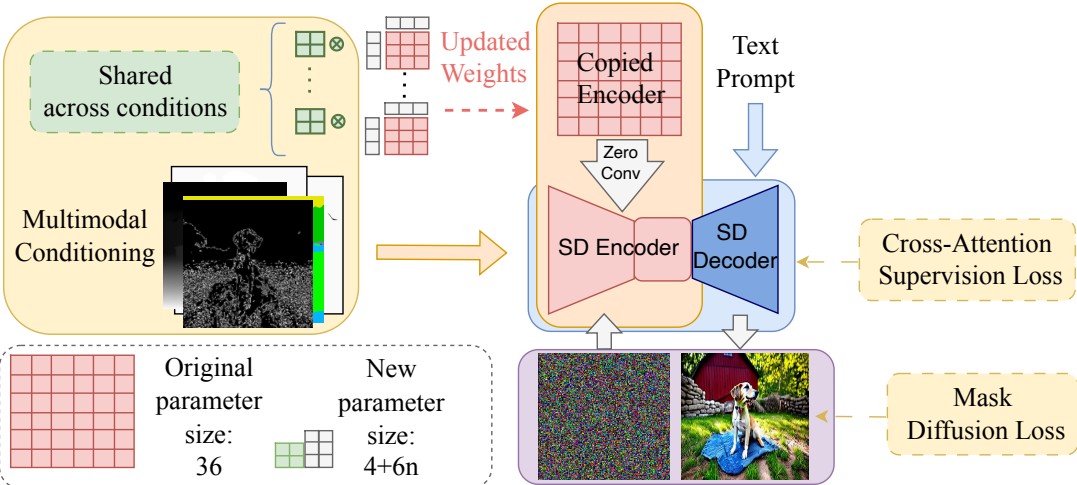

Figure 2: **Overview of FlexEControl**: a decomposed green matrix is shared across **different input conditions**, significantly enhancing the model's efficiency and preserving the image content. During training, we integrate two specialized loss functions to enable flexible control and to adeptly manage conflicting conditions. In the example depicted here, the new parameter size is efficiently condensed to $4 + 6n$, where $n$ denotes the number of decomposed matrix pairs.

2021a) and then low-rank decomposition over the updated weights of the copied Stable Diffusion encoder. To enhance the control from language and different input conditions, we propose a new training strategy with two newly designed loss functions. The details are shown in the sequel.

### 2.1 Preliminary

We use Stable Diffusion 1.5 (Rombach et al., 2022) in our experiments. This model falls under the category of Latent Diffusion Models (LDM) that encode input images $x$ into a latent representation $z$ via an encoder $\mathcal{E}$, such that $z = \mathcal{E}(x)$, and subsequently carry out the denoising process within the latent space $\mathcal{Z}$. An LDM is trained with a denoising objective as follows:

$$\mathcal{L}_{\text{ldm}} = \mathbb{E}_{z,c,e,t} \left[ \|\hat{\epsilon}_\theta(z_t \mid c, t) - \epsilon\|^2 \right] \tag{1}$$

where $(z, c)$ constitute data-conditioning pairs (comprising image latents and text embeddings), $\epsilon \sim \mathcal{N}(0, I)$, $t \sim \text{Uniform}(1, T)$, and $\theta$ denotes the model parameters.

### 2.2 Efficient Training for Controllable Text-to-Image (T2I) Generation

Our approach is motivated by empirical evidence that Kronecker Decomposition (Zhang et al., 2021a) effectively preserves critical weight information. We employ this technique to encapsulate the shared relational structures among different input conditions. Our hypothesis posits that by amalgamating diverse conditions with a common set of weights, data utilization can be optimized and training efficiency can be improved. We focus on decomposing and fine-tuning only the cross-attention weight matrices within the U-Net (Ronneberger et al., 2015) of the diffusion model, where recent works (Kumari et al., 2023) show their dominance when customizing the diffusion model. As depicted in Figure 2, the copied encoder from the Stable Diffusion will accept conditional input from different modalities. During training, we posit that these modalities, being transformations of the same underlying image, share common information. Consequently, we hypothesize that the updated weights of this copied encoder, $\Delta \boldsymbol{W}$, can be efficiently adapted within a shared decomposed low-rank subspace. This leads to:

$$\Delta \boldsymbol{W} = \sum_{i=1}^{n} \boldsymbol{H_i} \otimes \left( u_i v_i^\top \right) \tag{2}$$

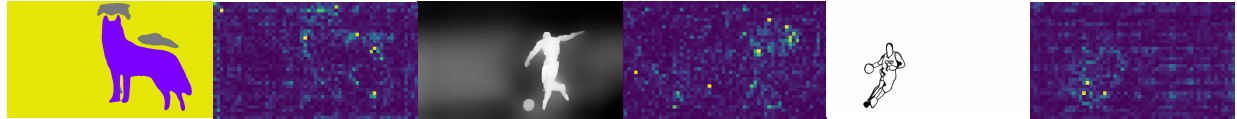

Figure 3: Visualization of decomposed shared "slow" weights (right image) for a single condition case where the input conditions (left image) are the segmentation, depth, and sketch maps, with the text prompt `Dog, Soccer player, Basketball player`. We averaged the decomposed shared weights from the last cross-attention block across all attention heads in Stable Diffusion. The results demonstrate the content-preserving properties of Kronecker Decomposition, where the "slow" weights effectively share semantic components.

with $n$ is the number of decomposed matrices, $u_i \in \mathbb{R}^{\frac{k}{n} \times r}$ and $v_i \in \mathbb{R}^{r \times \frac{d}{n}}$, where $r$ is the rank of the matrix which is a small number, $\boldsymbol{H_i}$ are the decomposed learnable matrices shared across different conditions, and $\otimes$ is the Kronecker product operation. The low-rank decomposition ensures a consistent low-rank representation strategy. This approach substantially saves trainable parameters, allowing efficient fine-tuning over the downstream text-to-image generation tasks.

The intuition for why Kronecker decomposition works for finetuning partially is partly rooted in the findings of Zhang et al. (2021a); Mahabadi et al. (2021); He et al. (2022). These studies highlight how the model weights can be broken down into a series of matrix products and thereby save parameter space. As shown in Figure 2, the original weights is 6x6, then decomposed into a series of matrix products. When adapting the training approach based on the decomposition to controllable T2I, the key lies in the shared weights, which, while being common across various conditions, retain most semantic information. The Kronecker Decomposition is known for its multiplicative rank property and content-preserving qualities. For instance, the shared "slow" weights (Wen et al., 2020) of an image, combined with another set of "fast" low-rank weights, can preserve the original image's distribution without a loss in semantic integrity, as illustrated in Figure 3. This observation implies that updating the slow weights is crucial for adapting to diverse conditions. Following this insight, it becomes logical to learn a set of condition-shared decomposed weights in each layer, ensuring that these weights remain consistent across different scenarios. The data utilization and parameter efficiency is also improved.

## 2.3 Enhanced Training for Conditional Inputs

We then discuss how to improve the control under multiple input conditions of varying modalities with the efficient training approach.

**Dataset Augmentation with Text Parsing and Segmentation** To optimize the model for scenarios involving multiple homogeneous (same-type) conditional inputs, we initially augment our dataset. We utilize a large language model (`gpt-3.5-turbo`) to parse texts in prompts containing multiple object entities. The parsing query is structured as: `Given a sentence, analyze the objects in this sentence, give me the objects if there are multiple.` Following this, we apply CLIPSeg (Lüddecke and Ecker, 2022) (`clipseg-rd64-refined` version) to segment corresponding regions in the images, allowing us to divide structural conditions into separate sub-feature maps tailored to the parsed objects. These segmentation masks are selectively used to augment the dataset, specifically when there is a clear, single mask for each identified object. This selective approach helps maintain the robustness of the dataset and enhances training performance.

**Cross-Attention Supervision Loss** For each identified segment, we calculate a unified attention map, $\boldsymbol{A}_i$, averaging attention across layers and relevant $N$ text tokens:

$$\boldsymbol{A}_i = \frac{1}{L} \sum_{l=1}^{L} \sum_{i=1}^{N} [\![ T_i \in \mathcal{T}_j ]\!] \mathbf{CA}_i^l, \tag{3}$$

where $[\![\cdot]\!]$ is the Iverson bracket, $\mathbf{CA}_i^l$ is the cross-attention map for token $i$ in layer $l$, and $\mathcal{T}_j$ denotes the set of tokens associated with the $j$-th segment.

The model is trained to predict noise for image-text pairs concatenated based on the parsed and segmented results. An additional loss term, designed to ensure focused reconstruction in areas relevant to each text-derived concept, is introduced. This loss is calculated as the Mean Squared Error (MSE) deviation from predefined masks corresponding to the segmented regions:

$$\mathcal{L}_{\mathrm{ca}} = \mathbb{E}_{z,t}\left[\left\|\boldsymbol{A}_i(v_i, z_t) - M_i\right\|_2^2\right], \tag{4}$$

where $\boldsymbol{A}_i(v_i, z_t)$ is the cross-attention map between token $v_i$ and noisy latent $z_t$, and $M_i$ represents the mask for the $i$-th segment, which is derived from the segmented regions in our augmented dataset and appropriately resized to match the dimensions of the cross-attention maps.

**Masked Diffusion Loss**  To ensure fidelity to the specified conditions, we apply a condition-selective diffusion loss that concentrates the denoising effort on conceptually significant regions. This focused loss function is applied solely to pixels within the regions delineated by the concept masks, which are derived from the non-zero features of the input structural conditions. Specifically, the masks are binary where non-zero feature areas are assigned a value of one, and areas lacking features are set to zero. Because of the sparsity of pose features for this condition, we use the all-ones mask. These masks serve to underscore the regions referenced in the corresponding text prompts:

$$\mathcal{L}_{\mathrm{mask}} = \mathbb{E}_{z,\epsilon,t}\left[\left\|(\epsilon - \epsilon_\theta(z_t, t)) \odot M\right\|_2^2\right], \tag{5}$$

where $M$ represents the union of binary mask obtained from input conditions, $z_t$ denotes the noisy latent at timestep $t$, $\epsilon$ the injected noise, and $\epsilon_\theta$ the estimated noise from the denoising network (U-Net).

The total loss function employed is:

$$\mathcal{L}_{\mathrm{total}} = \mathcal{L}_{\mathrm{ldm}} + \lambda_{\mathrm{ca}}\mathcal{L}_{\mathrm{ca}} + \lambda_{\mathrm{mask}}\mathcal{L}_{\mathrm{mask}}, \tag{6}$$

with $\lambda_{\mathrm{rec}}$ and $\lambda_{\mathrm{attn}}$ set to 0.01. The integration of $\mathcal{L}_{\mathrm{ca}}$ and $\mathcal{L}_{\mathrm{mask}}$ ensure the model will focus at reconstructing the conditional region and attend to guided regions during generation.

## 3 Experiments

### 3.1 Datasets

 In pursuit of our objective of achieving controlled Text-to-Image (T2I) generation, we employed the **LAION improved_aesthetics_6plus** (Schuhmann et al., 2022) dataset for our model training. Specifically, we meticulously curated a subset comprising 5,082,236 instances, undertaking the elimination of duplicates and applying filters based on criteria such as resolution and NSFW score. Given the targeted nature of our controlled generation tasks, the assembly of training data involved considerations of additional input conditions, specifically edge maps, sketch maps, depth maps, segmentation maps, and pose maps. The extraction of features from these maps adhered to the methodology expounded in(Zhang and Agrawala, 2023).

### 3.2 Experimental Setup

**Structural Input Condition Extraction**  We start from the processing of various local conditions used in our experiments. To facilitate a comprehensive evaluation, we have incorporated a diverse range of structural conditions. These conditions include edge maps (Canny, 1986; Xie and Tu, 2015; Gu et al., 2022a), sketch maps (Simo-Serra et al., 2016), pose information (Cao et al., 2017), depth maps (Ranftl et al., 2020), and segmentation maps (Xiao et al., 2018), each extracted using specialized techniques. These conditions are crucial for guiding the text-to-image generation process, enabling FlexEControl to produce images that are both visually appealing and semantically aligned with the text prompts and structural inputs. The additional details for extracting those conditions are given in the Appendix.

Table 1: **Text-to-image generation efficiency comparison**: FlexEControl shows substantial reductions in memory cost, trainable parameters, and training time, highlighting its improved training efficiency with the same model architecture. Training times are averaged over three runs up to 400 iterations for consistency.

| Models | Memory Cost ↓ | # Params. ↓ | Training Time ↓ |
|---|---|---|---|
| Uni-ControlNet (Zhao et al., 2023) | 20.47GB | 1271M | 5.69 ± 1.33s/it |
| LoRA (Hu et al., 2021) | 17.84GB | 1074M | 3.97 ± 1.27 s/it |
| PHM (Zhang et al., 2021a) | 15.08GB | 819M | 3.90 ± 2.01 s/it |
| FlexEControl (**ours**) | **14.33GB** | **750M** | **2.15 ± 1.42 s/it** |

Table 2: **Quantitative evaluation of controllability and image quality** for single structural conditional inputs. FlexEControl performs overall better while maintaining much improved efficiency.

| Models | Canny (SSIM)↑ | MLSD (SSIM)↑ | HED (SSIM)↑ | Sketch (SSIM)↑ | Depth (MSE)↓ | Segmentation (mIoU)↑ | Poses (mAP)↑ | FID↓ | CLIP Score↑ |
|---|---|---|---|---|---|---|---|---|---|
| T2IAdapter (Mou et al., 2023) | 0.4480 | - | - | 0.5241 | 90.01 | 0.6983 | **0.3156** | 27.80 | 0.4957 |
| ControlNet (Zhang and Agrawala, 2023) | 0.4989 | 0.6172 | 0.4990 | **0.6013** | **89.08** | 0.7481 | 0.2024 | 27.62 | 0.4931 |
| Uni-Control (Qin et al., 2023) | 0.4977 | 0.6374 | 0.4885 | 0.5509 | 90.04 | 0.7143 | 0.2083 | 27.80 | 0.4899 |
| Uni-ControlNet (Zhao et al., 2023) | 0.4910 | 0.6083 | 0.4715 | 0.5901 | 90.17 | 0.7084 | 0.2125 | 27.74 | 0.4890 |
| PHM (Zhang et al., 2021a) | 0.4365 | 0.5712 | 0.4633 | 0.4878 | 91.38 | 0.5534 | 0.1664 | 27.91 | 0.4961 |
| LoRA (Hu et al., 2021) | 0.4497 | 0.6381 | **0.5043** | 0.5097 | 89.09 | 0.5480 | 0.1538 | 27.99 | 0.4832 |
| FlexEControl (**ours**) | **0.4990** | **0.6385** | 0.5041 | 0.5518 | 90.93 | **0.7496** | 0.2093 | **27.55** | **0.4963** |

**Evaluation Metrics**   We employ a comprehensive benchmark suite of metrics including mIoU (Rezatofighi et al., 2019), SSIM (Wang et al., 2004), mAP, MSE, FID (Heusel et al., 2017), and CLIP Score (Hessel et al., 2021; Radford et al., 2021) [1].

**Baselines**   In our comparative evaluation, we assess T2I-Adapter (Mou et al., 2023), PHM (Zhang et al., 2021a), Uni-ControlNet (Zhao et al., 2023), and LoRA (Hu et al., 2021). The implementation details are given in the Appendix.

**Implementation Details**   In accordance with the configuration employed in Uni-ControlNet, we utilized Stable Diffusion 1.5 [2] as the foundational model. Our model underwent training for a singular epoch, employing the AdamW optimizer (Kingma and Ba, 2014) with a learning rate set at $10^{-5}$. Throughout all experimental iterations, we standardized the dimensions of input and conditional images to $512 \times 512$. The fine-tuning process was executed on P3 AWS EC2 instances equipped with 64 NVIDIA V100 GPUs.

For baseline implementations, we compare FlexEControl with T2I-Adapter (Mou et al., 2023), PHM (Zhang et al., 2021a), Uni-ControlNet (Zhao et al., 2023), and LoRA (Hu et al., 2021) where we implement LoRA and PHM layers over the trainable modules in Uni-ControlNet interms of generated image quality and controllability. The rank of LoRA is set to 4. For PHM (Zhang et al., 2021a), we implement it by performing Kronecker decomposition and share weights across different layer, with the number of decomposed matrix being 4.

For quantitative assessment, a subset comprising 10,000 high-quality images from the **LAION improved_aesthetics_6.5plus** dataset was utilized. The resizing of input conditions to $512 \times 512$ was conducted during the inference process.

### 3.3   Quantitative Results

Table 1 highlights FlexEControl's superior efficiency compared to Uni-ControlNet. It achieves a 30% reduction in memory cost, lowers trainable parameters by 41% (from 1271M to 750M), and significantly reduces training time per iteration from 5.69s to 2.15s.

---

[1]https://github.com/jmhessel/clipscore
[2]https://huggingface.co/runwayml/stable-diffusion-v1-5

Table 3: **Quantitative evaluation of controllability and image quality** on FlexEControl along with its variants and Uni-ControlNet. For Uni-ControlNet, we implement multiple conditioning by adding two homogeneous conditional images after passing them through the feature extractor.

| | Models | Canny (SSIM)↑ | MLSD (SSIM)↑ | HED (SSIM)↑ | Sketch (SSIM)↑ | Depth (MSE)↓ | Segmentation (mIoU)↑ | Poses (mAP)↑ | FID↓ | CLIP Score↑ |
|---|---|---|---|---|---|---|---|---|---|---|
| Single Conditioning | Uni-ControlNet | 0.3268 | 0.4097 | 0.3177 | 0.4096 | 98.80 | 0.4075 | **0.1433** | 29.43 | 0.4844 |
| | FlexEControl (w/o $L_{ca}$) | 0.3698 | 0.4905 | 0.3870 | 0.4855 | 94.90 | 0.4449 | 0.1432 | 28.03 | 0.4874 |
| | FlexEControl (w/o $L_{mask}$) | 0.3701 | 0.4894 | 0.3805 | **0.4879** | **94.30** | 0.4418 | 0.1432 | 28.19 | 0.4570 |
| | FlexEControl | **0.3711** | **0.4920** | **0.3871** | 0.4869 | 94.83 | **0.4479** | 0.1432 | **28.03** | **0.4877** |
| Multiple Conditioning | Uni-ControlNet | 0.3078 | 0.3962 | 0.3054 | 0.3871 | 98.84 | 0.3981 | 0.1393 | 28.75 | 0.4828 |
| | FlexEControl (w/o $L_{ca}$) | 0.3642 | 0.4901 | 0.3704 | 0.4815 | 94.95 | 0.4368 | 0.1405 | 28.50 | 0.4870 |
| | FlexEControl (w/o $L_{mask}$) | 0.3666 | 0.4834 | 0.3712 | 0.4831 | 94.89 | 0.4400 | 0.1406 | 28.68 | 0.4542 |
| | FlexEControl | **0.3690** | **0.4915** | **0.3784** | **0.4849** | **92.90** | **0.4429** | **0.1411** | **28.24** | **0.4873** |

Table 2 provides a comprehensive comparison of FlexEControl's performance against Uni-ControlNet and T2IAdapter across diverse input conditions. After training on a dataset of 5M text-image pairs, FlexEControl demonstrates better, if not superior, performance metrics compared to Uni-ControlNet and T2IAdapter. Note that Uni-ControlNet is trained on a much larger dataset (10M text-image pairs from the LAION dataset). Although there is a marginal decrease in SSIM scores for sketch maps and mAP scores for poses, FlexEControl excels in other metrics, notably surpassing Uni-ControlNet and T2IAdapter. This underscores our method's proficiency in enhancing efficiency and elevating overall quality and accuracy in controllable text-to-image generation tasks.

To validate FlexEControl's effectiveness in handling multiple structural conditions, we compared it with Uni-ControlNet through human evaluations. Two scenarios were considered: multiple homogeneous input conditions (300 images, each generated with 2 canny edge maps) and multiple heterogeneous input conditions (500 images, each generated with 2 randomly selected conditions). Results, summarized in Table 4, reveal that FlexEControl was preferred by 64.00% of annotators, significantly outperforming Uni-ControlNet (23.67%). This underscores FlexEControl's proficiency with complex, homogeneous inputs. Additionally, FlexEControl demonstrated superior alignment with input conditions (67.33%) compared to Uni-ControlNet (23.00%). In scenarios with random heterogeneous conditions, FlexEControl was preferred for overall quality and alignment over Uni-ControlNet.

In addition to our primary comparisons, we conducted an additional quantitative evaluation of FlexEControl and Uni-ControlNet. This evaluation focused on assessing image quality under scenarios involving multiple conditions from both the homogeneous and heterogeneous modalities. The findings of this evaluation are summarized in Table 5. FlexEControl consistently outperforms Uni-ControlNet in both categories, demonstrating lower FID scores for better image quality and higher CLIP scores for improved alignment with text prompts.

### 3.3.1 Ablation Studies

To substantiate the efficacy of FlexEControl in enhancing training efficiency while upholding commendable model performance, and to ensure a fair comparison, an ablation study was conducted by training models on an identical dataset. We trained FlexEControl along its variants and Uni-ControlNet on a subset of 100,000 training samples from **LAION improved_aesthetics_6plus**. When trained with the identical data, FlexEControl performs better than Uni-ControlNet. The outcomes are presented in Table 3. Evidently, FlexEControl exhibits substantial improvements over Uni-ControlNet when trained on the same dataset. This underscores the effectiveness of our approach in optimizing data utilization, concurrently diminishing computational costs, and enhancing efficiency in the text-to-image generation process.

We also study the impact of $\lambda_{ca}$ and $\lambda_{mask}$ trained on the subset of 100,000 samples from **LAION improved_aesthetics_6plus** for 6,000 steps. We evaluated the score on SSIM of canny edge maps and mIoU of segmentation maps, results are shown in Figure 4. As observed, FlexEControl achieves optimal performance when both $\lambda_{ca} = 0.01$ and $\lambda_{mask} = 0.01$. Additionally, as indicated in Table 3, FlexEControl outperforms the configurations where either $\lambda_{ca}$ or $\lambda_{mask}$ is set to zero, demonstrating the importance of incorporating both the cross-attention supervision loss and the masked diffusion loss. However, higher values

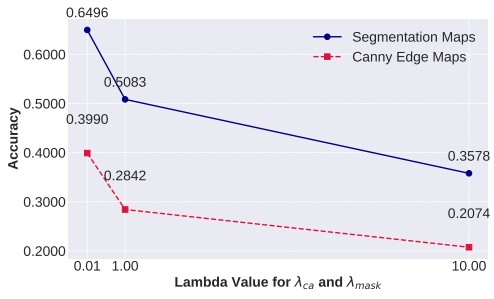
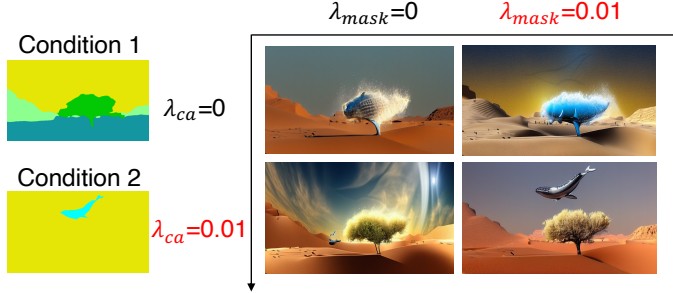

(a) FlexEControl performs the best when both $\lambda_{\text{ca}} = 0.01$ and $\lambda_{\text{mask}} = 0.01$.

(b) Qualitative comparison. The text prompt is `A mechanical whale flying over a desert that has a tree.`

Figure 4: Quantitative and qualitative comparison showing the effect of excluding cross-attention supervision loss and masked diffusion loss.

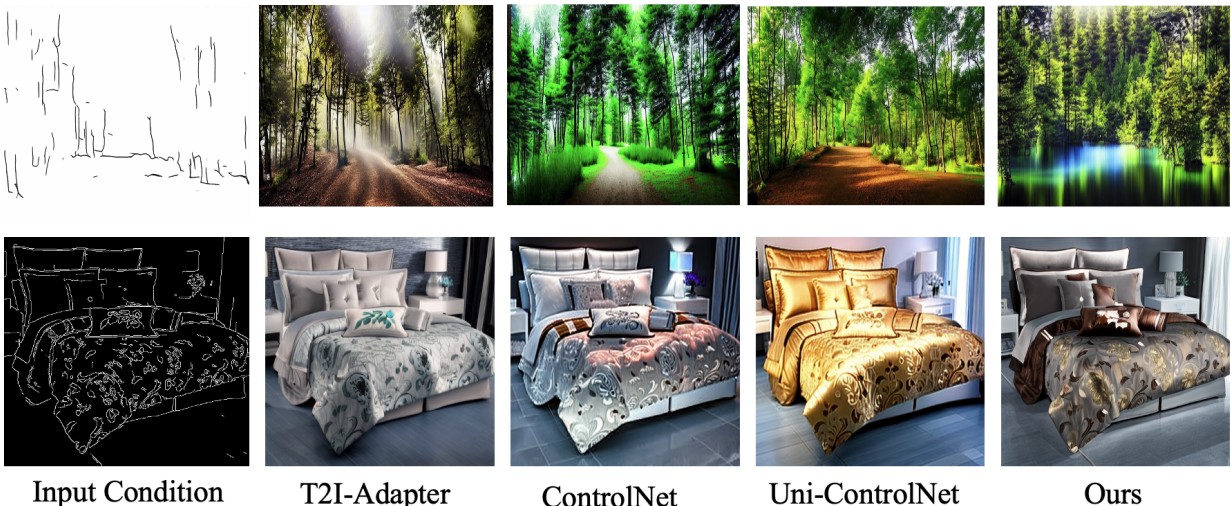

Figure 5: Qualitative comparison of FlexEControl and existing controllable diffusion models with single condition. Text prompt: `A bed.` The image quality of FlexEControl is comparable to existing methods and Uni-ControlNet + LoRA, while FlexEControl has much more efficiency.

of $\lambda_{\text{ca}}$ and $\lambda_{\text{mask}}$ could lead to instability during training, causing the model to focus excessively on local conditions, which results in worse performance. We also show the qualitative comparison of $\lambda_{\text{ca}}$ and $\lambda_{\text{mask}}$ in Figure 4. Without the cross-attention loss, the concept of the whale tends to merge with the tree, causing attribute leakage and misblending issues. Without the masked diffusion loss, the model is more likely to generate the whale outside the intended region, leading to blurred regions and a lack of focus within the maps.

### 3.3.2 Additional Results on Stable Diffusion 2

In our efforts to explore the versatility and adaptability of FlexEControl, we conducted additional experiments using the Stable Diffusion 2.1 model, available at Hugging Face's Model Hub. The results from these experiments are depicted in Table 6. FlexEControl can leverage the advancements in Stable Diffusion 2.1 to achieve even better performance in text-to-image generation tasks. For the sake of a fair comparison in the main paper, we conduct experiments using Stable Diffusion 1.5 model.

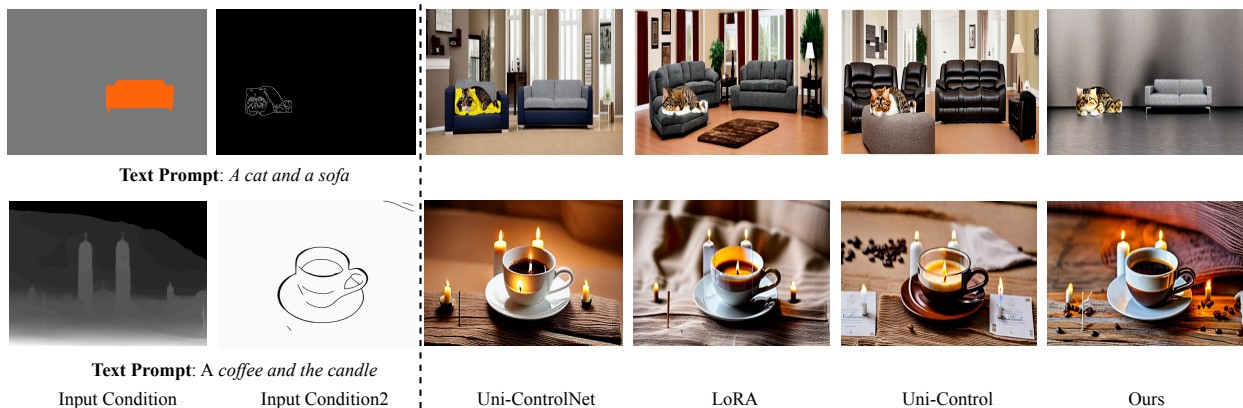

**Text Prompt**: *A cat and a sofa*

**Text Prompt**: A *coffee and the candle*

Input Condition      Input Condition2      Uni-ControlNet      LoRA      Uni-Control      Ours

Figure 6: Qualitative comparison of FlexEControl and existing controllable diffusion models with multiple heterogeneous conditions. First row: FlexEControl effectively integrates both the segmentation and edge maps to generate a coherent image while Uni-ControlNet and LoRA miss the segmentation map and Uni-Control generates a messy image. Additionally, the cats generated by Uni-ControlNet, LoRA, and Uni-Control lack clear feline characteristics and does not align with the edge map. Second row: The input condition types are one depth map and one sketch map. FlexEControl can do more faithful generation while all three others generate the candle in the coffee.

Table 4: Human evaluation of FlexEControl and Uni-ControlNet under homogenous and heterogeneous structural conditions, assessing both human preference and condition alignment. "Win" indicates FlexEControl's preference, "Tie" denotes equivalence, and "Lose" indicates Uni-ControlNet's preference. Results indicate that under homogeneous conditions, FlexEControl outperforms Uni-ControlNet in both human preference and condition alignment.

| Condition Type | Metric | Win | Tie | Lose |
|---|---|---|---|---|
| Homogeneous | Human Preference (%) | **64.00** | 12.33 | 23.67 |
| | Condition Alignment (%) | **67.33** | 9.67 | 23.00 |
| Heterogeneous | Human Preference (%) | **9.80** | 87.40 | 2.80 |
| | Condition Alignment (%) | **6.60** | 89.49 | 4.00 |

Table 5: Quantitative evaluation of controllability and image quality in scenarios with multiple conditions from heterogeneous and homogeneous modalities for FlexEControl and Uni-ControlNet. The 'heterogeneous' category averages the performance across one Canny condition combined with six other different modalities. The 'homogeneous' category represents the average performance across seven identical modalities (three inputs).

| Condition Type | Baseline | FID↓ | CLIP Score↑ |
|---|---|---|---|
| Heterogeneous | Uni-ControlNet | 27.81 | 0.4869 |
| | FlexEControl | **27.47** | **0.4981** |
| Homogeneous | Uni-ControlNet | 28.98 | 0.4858 |
| | FlexEControl | **27.65** | **0.4932** |

## 3.4 Qualitative Results

We present qualitative results of our FlexEControl under three different settings: single input condition, multiple heterogeneous conditions, and multiple homogeneous conditions, illustrated in Figure 5, Figure 6, and Figure 7, respectively. The results indicate that FlexEControl is comparable to baseline models when

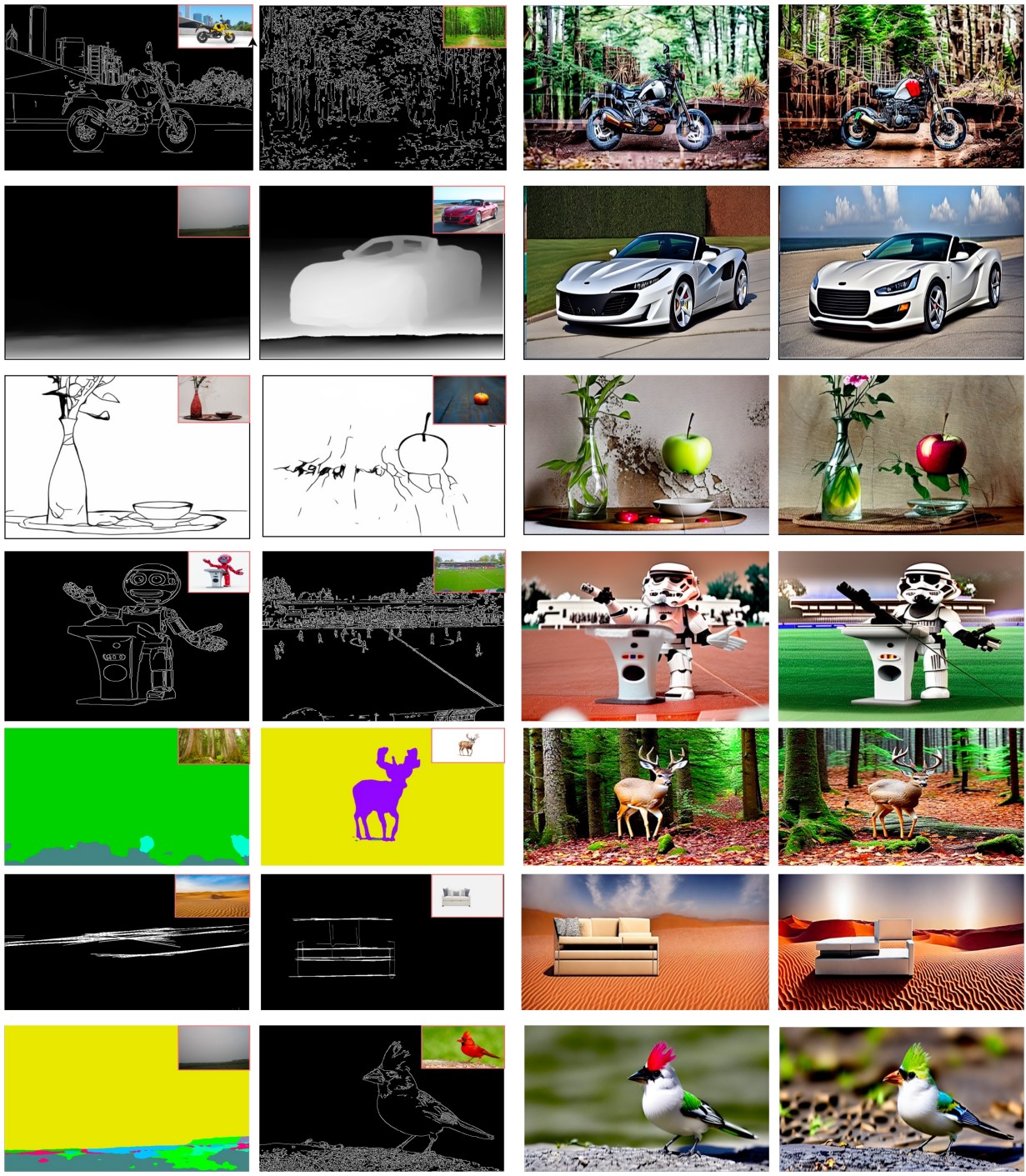

Figure 7: Qualitative performance of FlexEControl when conditioning on diverse compositions of multiple modalities. Each row in the figure corresponds to a unique type of condition, with the text prompts and conditions as follows: (first row) two canny edge maps with the prompt `A motorcycle in the forest`, (second row) two depth maps for `A car`, (third row) two sketch maps depicting `A vase with a green apple`, (fourth row) two canny edge maps for `Stormtrooper's lecture at the football field`, (fifth row) two segmentation maps visualizing `A deer in the forests`, (sixth row) two MLSD edge maps for `A sofa in a desert`, and (seventh row) one segmentation map and one edge map for `A bird`. These examples illustrate the robust capability of FlexEControl to effectively utilize multiple multimodal conditions, generating images that are not only visually compelling but also faithfully aligned with the given textual descriptions and input conditions.

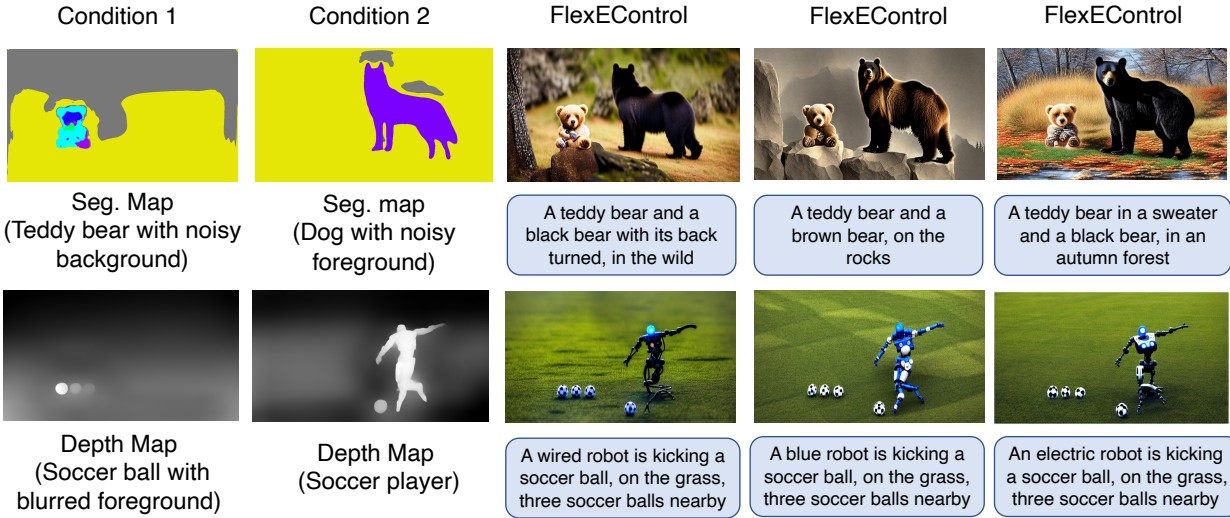

Figure 8: Qualitative results on multimodal control for generating multiple foregrounds.

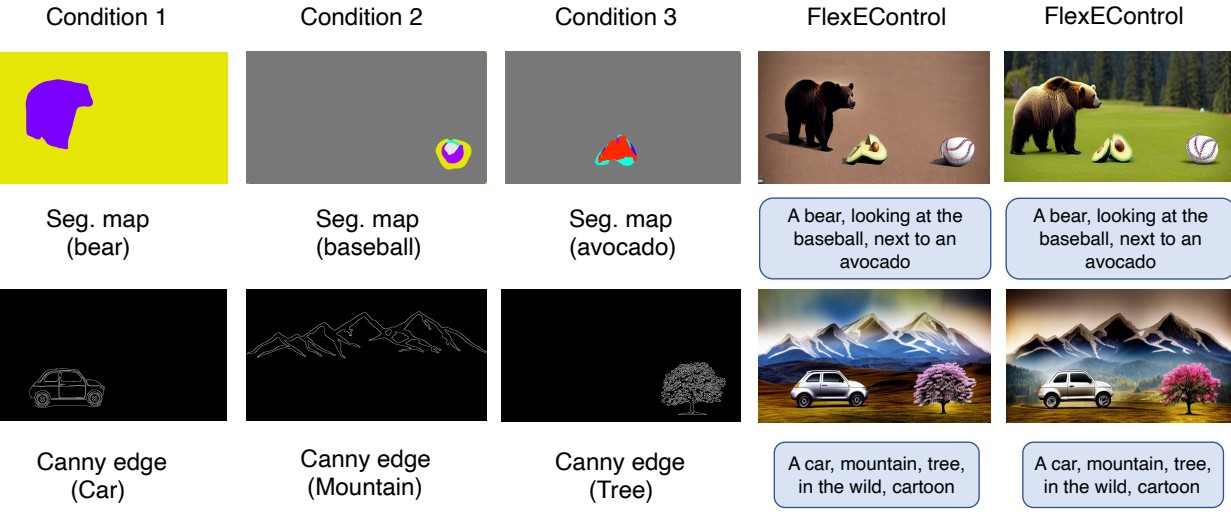

Figure 9: Qualitative results on multimodal control using three homogeneous condition images

a single condition is input. However, with multiple conditions, FlexEControl consistently and noticeably outperforms other models. Particularly, under multiple homogeneous conditions, FlexEControl excels in generating overall higher quality images that align more closely with the input conditions, surpassing other models.

We also shown in Figure 8 the qualitative results of generating multiple foregrounds.. In the teddy bear and dog condition case, the teddy bear has a noisy background, and the dog image has a noisy foreground, while the prompt asks the model to generate a teddy bear and a bear. As shown in the results, FlexEControl effectively handles the noisy and conflicting semantic information, successfully following the prompt to generate an image with two foregrounds: a teddy bear and a bear, even though the bear condition comes from a dog image. In the robot case, the depth map for the soccer ball is blurred, and the player given is a human soccer player, yet the model still follows the prompt to generate robot soccer player images with three additional soccer balls. These examples demonstrate the strong controllability of FlexEControl.

Furthermore, we show in Figure 9 the qualitative results for FlexEControl on mulimodal control using three homogeneous conditiona images. Further highlighting the versatility and effectiveness of FlexEControl in

Table 6: Quantitative evaluation of controllability and image quality trained on a subset of 100,000 samples. Human poses are evaluated solely within portrait images.

| Models | Canny (SSIM)↑ | MLSD (SSIM)↑ | HED (SSIM)↑ | Sketch (SSIM)↑ | Depth (MSE)↓ | Segmentation (mIoU)↑ | Poses (mAP)↑ | FID↓ | CLIP Score↑ |
|---|---|---|---|---|---|---|---|---|---|
| FlexEControl | 0.3711 | 0.4920 | 0.3871 | 0.4869 | 94.83 | 0.4479 | 0.1432 | 28.03 | 0.4877 |
| FlexEControl-SD 2.1 | **0.3891** | **0.5273** | **0.4077** | **0.4960** | **93.58** | **0.4490** | **0.1562** | **25.08** | **0.5833** |

| Condition 1 | Condition 2 | Ours | Condition 1 | Condition 2 | Ours |
|---|---|---|---|---|---|

Figure 10: **Failure cases**. Failure cases in generating human images (*left*): The text prompt is: `a basketball player with a helicopter.` Due to limitations in the Stable Diffusion backbone, FlexEControl fails to generate a correct basketball player, instead producing an image where the basketball player's face is replaced with a basketball. Failure cases in generating images with a weak text prompt for the background (*right*): The text prompt is `a car is parking`, which provides little information about the background. The background segmentation map is complex, and the foreground uses a strong canny edge guidance. Consequently, the generated image shows a weak effect of the segmentation, although the foreground is generated accurately.

handling multiple conditions. Additionally, we showcase the extensibility of FlexEControl in controllable video generation. The results are presented in Figure 12 and Figure 13 in the Appendix, where results for providing one condition and multiple conditions are demonstrated.

### 3.5 Limitations

Although FlexEControl achieves strong performance, it shares the inherent limitations of diffusion-based image generation models. The LAION dataset exhibits certain biases, which can lead to suboptimal performance in specific scenarios. FlexEControl could benefit from a more robust and strong pretrained text-to-image generation backbone and could be further improved with access to better open-source datasets that mitigate the risk of generating biased, toxic, sexualized, or other harmful content. Some failure cases and analyses where FlexEControl struggled are illustrated in Figure 10.

## 4 Related Work

FlexEControl is an instance of efficient training and controllable text-to-image generation. Here, we overview modeling efforts in the subset of efficient training towards reducing parameters and memory cost and controllable T2I.

### 4.1 Efficient Training

Prior work has proposed efficient training methodologies both for pretraining and fine-tuning. These methods have established their efficacy across an array of language and vision tasks. One of these explored strategies is Prompt Tuning (Lester et al., 2021), where trainable prompt tokens are appended to pretrained models (Schick and Schütze, 2020; Ju et al., 2021; Jia et al., 2022). These tokens can be added exclusively to input embeddings or to all intermediate layers (Li and Liang, 2021), allowing for nuanced model control and performance optimization. Low-Rank Adaptation (LoRA) (Hu et al., 2021) is another innovative approach that introduces trainable rank decomposition matrices for the parameters of each layer. LoRA has exhibited promising fine-tuning ability on large generative models, indicating its potential for broader application. Furthermore, the use of Adapters inserts lightweight adaptation modules into each layer of a pretrained transformer (Houlsby et al., 2019; Rücklé et al., 2021). This method has been successfully extended across various setups (Zhang et al.,

2021b; Gao et al., 2021; Mou et al., 2023), demonstrating its adaptability and practicality. Other approaches including post-training model compression (Fang et al., 2023) facilitate the transition from a fully optimized model to a compressed version – either sparse (Frantar and Alistarh, 2023), quantized (Li et al., 2023a; Gu et al., 2022b), or both. This methodology was particularly helpful for parameter quantization (Dettmers et al., 2023). Different from these methodologies, FlexEControl puts forth a new unified strategy that aims to enhance the efficient training of text-to-image diffusion models through the leverage of low-rank structure. Yeh et al. (2023) and Marjit et al. (2024) explore the use of Kronecker Product for tuning T2I models. These approaches focus on enhancing control and efficiency, particularly in subject-driven or style-preserving generation tasks. FlexEControl is the first to leverage it for multimodal conditioning, leading to enhanced flexibility and efficiency in handling diverse input modalities. FlexEControl is also a general approach that can be applied to UniControl Qin et al. (2023) or other backbones.

## 4.2 Controllable Text-to-Image Generation

Recent developments in the text-to-image generation domain strives for more control over image generation, enabling more targeted, stable, and accurate visual outputs, several models like T2I-Adapter (Mou et al., 2023) and Composer (Huang et al., 2023) have emerged to enhance image generations following the semantic guidance of text prompts and multiple different structural conditional control. However, existing methods are struggling at dealing with multiple conditions from the same modalities, especially when they have conflicts, e.g. multiple segmentation maps and at the same time follow the guidance of text prompts; Recent studies also highlight challenges in controllable text-to-image generation (T2I), such as omission of objects in text prompts and mismatched attributes (Lee et al., 2023; Bakr et al., 2023), showing that current models are struggling at handling controls from different conditions. Towards these, the Attend-and-Excite method Chefer et al. (2023) refines attention regions to ensure distinct attention across separate image regions. ReCo Yang et al. (2023), GLIGEN Li et al. (2023b), and Layout-Guidance Chen et al. (2023) allow for image generation informed by bounding boxes and regional descriptions. Mo et al. (2024) offers a training-free approach to multimodal control. FlexEControl improves the model's controllability by proposing a new training strategy, distinguishing itself by targeting the flexibility and efficiency of multimodal control, especially in scenarios with conflicting conditions from the same or different modalities (e.g., multiple segmentation maps combined with text prompts).

## 5 Conclusion

In this work, we present FlexEControl, an approach designed to enhance both the flexibility and efficiency of controllable diffusion-based text-to-image generation. Our method introduces several key innovations: Dataset Augmentation with Text Parsing and Segmentation, Cross-Attention Supervision, and Masked Diffusion Loss, which together significantly improve the model's ability to handle diverse and conflicting multimodal inputs. Additionally, we propose an Efficient Training strategy that optimizes parameter, data, and memory efficiency without sacrificing performance or inference speed. We also demonstrate that sharing a common set of decomposed weights across different multimodal conditions via Kronecker Decomposition can further optimize parameter space and enhance efficiency. These findings suggest that FlexEControl can be readily adapted to other architectures, offering a scalable solution for future developments in text-to-image generation. Future work could explore more advanced decomposition techniques and their application to cutting-edge diffusion backbones or Diffusion Transformers (DiTs), aiming to further optimize model efficiency, complexity, and expressive power.

## Broader Impact Statement

While FlexEControl demonstrates promising results in efficient and controllable text-to-image generation, the ability of FlexEControl to generate realistic images based on textual descriptions raises ethical concerns, especially regarding the creation of misleading or deceptive content. It is imperative to establish guidelines and ethical standards for the use of such technology to prevent misuse in generating deepfakes or propagating false information.

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

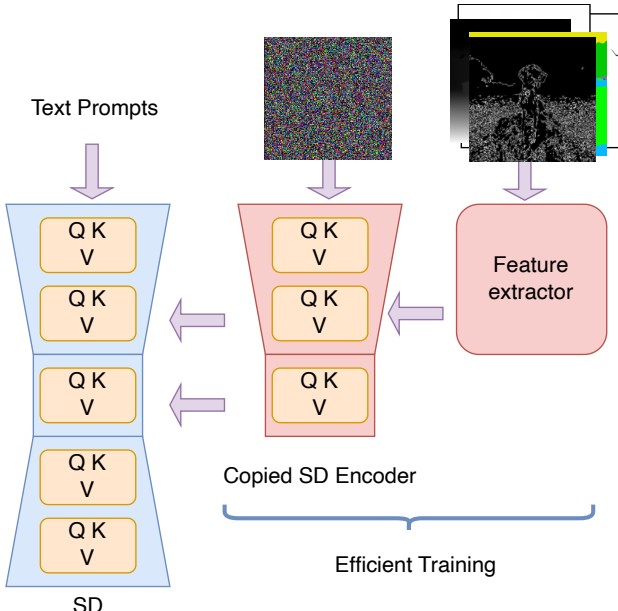

Figure 11: Detailed model architecture in FlexEControl. The Stable Diffusion part is fixed and others are trainable.

This Appendix is organized as follows:

- Appendix A contains our detailed model architectures;

- Appendix B contains additional implementation details;

- Appendix C contains additional results;

- Appendix D contains additional related works;

- Appendix E contains details for the human evaluation setup;

## A    Detailed Model Architecture

In ControlNet (Zhang and Agrawala, 2023) and Uni-ControlNet (Zhao et al., 2023), the weights of Stable Diffusion (SD) (Rombach et al., 2022) are fixed and the input conditions are fed into zero-convolutions and added back into the main Stable Diffusion backbone. Specifically, for Uni-ControlNet, they uses a multi-scale condition injection strategy that extracts features at different resolutions and uses them for condition injection referring to the implementation of Feature Denormalization (FDN):

$$
\begin{aligned}
\mathrm{FDN}\left(Z, c\right) &= \mathrm{norm}\left(Z\right) \cdot \left(1 + \Phi\left(\mathrm{zero}\left(h_r\left(c\right)\right)\right)\right) \\
&+ \Phi\left(\mathrm{zero}\left(h_r\left(c\right)\right)\right),
\end{aligned}
\tag{7}
$$

where $Z$ denotes noise features, $c$ denotes the input conditional features, $\Phi$ denotes learnable convolutional layers, and zero denotes zero convolutional layer. The zero convolutional layer contains weights initialized to zero. This ensures that during the initial stages of training, the model relies more on the knowledge from the backbone part, gradually adjusting these weights as training progresses. The use of such layers aids in preserving the architecture's original behavior while introducing structure-conditioned inputs. We use the similar model architecture while we perform efficient training proposed in the main paper. We show the model architecture in Figure 11.

# B    Additional Implementation Details

In this section, we provide further details about the implementation aspects of our approach.

## B.1    Additional Details of Structural Input Conditions Extraction

- **Edge Maps**: For generating edge maps, we utilized two distinct techniques:
    - Canny Edge Detector (Canny, 1986) - A widely used method for edge detection in images.
    - HED Boundary Extractor (Xie and Tu, 2015) - Holistically-Nested Edge Detection, an advanced technique for identifying object boundaries.
    - MLSD (Gu et al., 2022a) - A method particularly designed for detecting multi-scale line segments in images.

- **Sketch Maps**: We adopted a sketch extraction technique detailed in Simo-Serra et al. (2016) to convert images into their sketch representations.

- **Pose Information**: OpenPose (Cao et al., 2017) was employed to extract human pose information from images, which provides detailed body joint and keypoint information.

- **Depth Maps**: For depth estimation, we integrated Midas (Ranftl et al., 2020), a robust method for predicting depth information from single images.

- **Segmentation Maps**: Segmentation of images was performed using the method outlined in Xiao et al. (2018), which focuses on accurately segmenting various objects within an image.

Each of these conditions plays a crucial role in guiding the text-to-image generation process, helping FlexEControl to generate images that are not only visually appealing but also semantically aligned with the given text prompts and structural conditions.

## B.2    Additional Details of Evaluation Metrics

**mIoU (Rezatofighi et al., 2019):**   Mean Intersection over Union, a metric that quantifies the degree of overlap between predicted and actual segmentation maps.

**SSIM (Wang et al., 2004):**   Structural Similarity, a metric evaluating the structural similarity in generated outputs, applied to Canny edges, HED edges, MLSD edges, and sketches.

**mAP:**   Mean Average Precision, utilized for pose maps, measuring the precision of localization across multiple instances.

**MSE:**   Mean Squared Error, employed for depth maps, MSE quantifies the pixel-wise variance, providing an assessment of image fidelity.

**FID (Heusel et al., 2017):**   Fréchet Inception Distance, which serves as a metric to quantify the realism and diversity of the generated images. A lower FID value indicates higher quality and diversity of the output images.

**CLIP Score (Hessel et al., 2021; Radford et al., 2021):**   Employing CLIP Score, we gauge the semantic similarity between the generated images and the input text prompts.

# C    Additional Qualitative Results on Video Generation

In this section, we showcase the extensibility of FlexEControl in controllable video generation. The results are presented in Figure 12 and Figure 13, where results for providing one condition and multiple conditions are demonstrated.

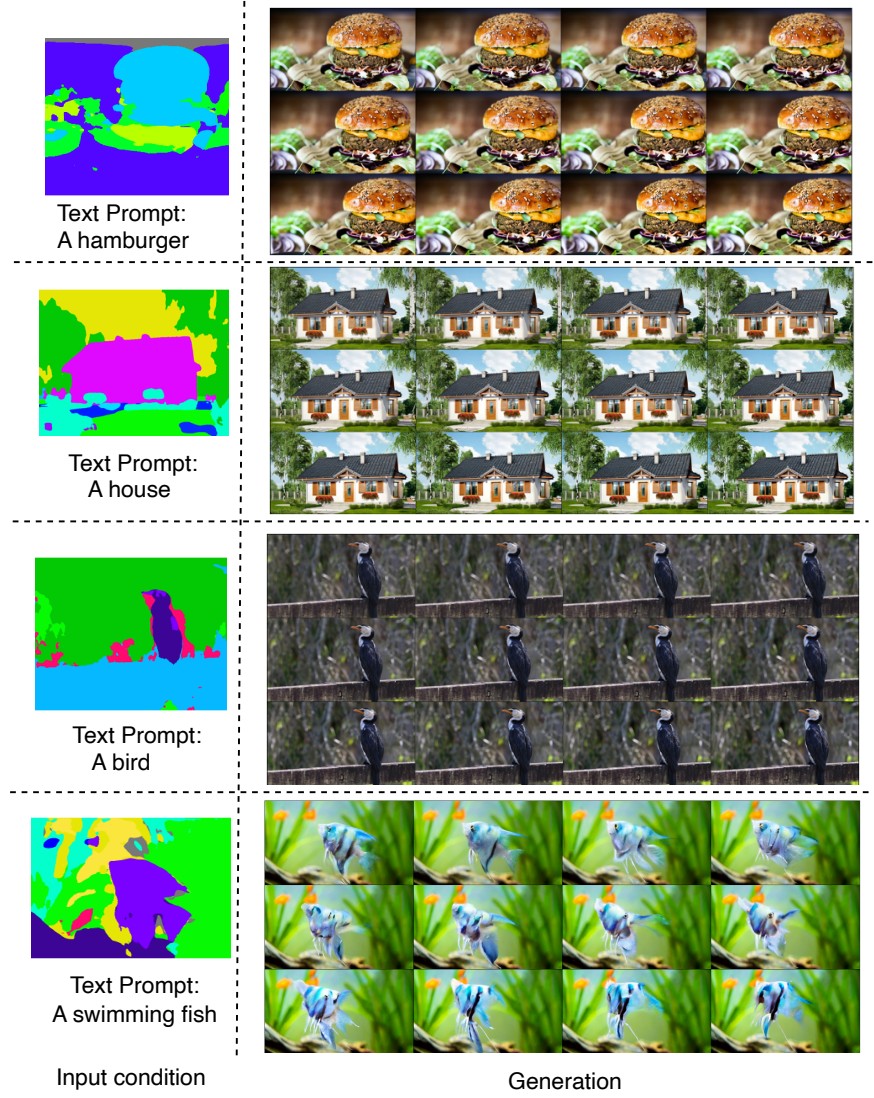

Figure 12: Results from FlexEControl on controllable text-to video generation (single condition).

# D   Additional Related Works

**Knowledge Distillation for Vision-and-Language Models**   Knowledge distillation (Gou et al., 2021), as detailed in prior research, offers a promising approach for enhancing the performance of a more streamlined "student" model by transferring knowledge from a more complex "teacher" model (Hinton et al., 2015; Sanh et al., 2019; Hu et al.; Gu et al., 2021; Li et al., 2021). The crux of this methodology lies in aligning the predictions of the student model with those of the teacher model. While a significant portion of existing knowledge distillation techniques leans towards employing pretrained teacher models (Tolstikhin et al., 2021), there has been a growing interest in online distillation methodologies (Wang and Jordan, 2021). In online distillation (Guo et al., 2020), multiple models are trained simultaneously, with their ensemble serving as the teacher. Our approach is reminiscent of online self-distillation, where a temporal and resolution ensemble of the student model operates as the teacher. This concept finds parallels in other domains, having been examined in semi-supervised learning (Peters et al., 2017), label noise learning (Bengio et al., 2010), and quite recently in contrastive learning (Chen et al., 2020). Our work on distillation for pretrained text-to-image generative diffusion models distinguishes our method from these preceding works.  (Salimans and Ho, 2022;

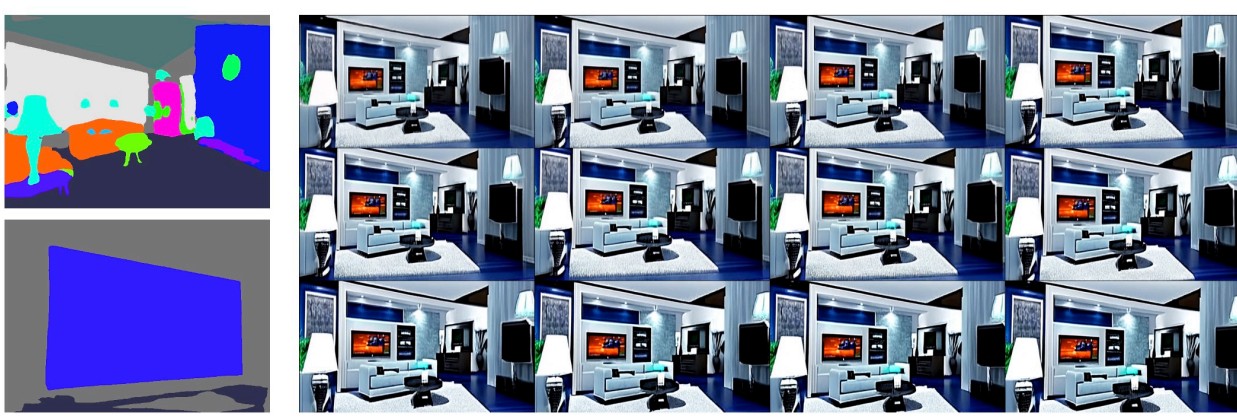

Text Prompt: A TV in the living room

Figure 13: FlexEControl on using multiple conditions for video generation.

You will see **two input images (edge maps) and a text prompt**, along with **two generated output images**.
The goal is to generate images that align with both the input images and the text prompt.

Your task is to evaluate the two generated images based on the following criteria:
[1] Alignment with Input: Which output image aligns better with the input conditions (edge maps)?
[2] Overall Preference: Considering all aspects, which output image do you prefer? This includes:
   a) Semantic relevance: Does the output image align well with the text prompt?
   b) Image quality: Is the output image of good visual quality?
   c) Coherence: Does the output image properly reflect the edges shown in the input images?

Question 1: Which output image aligns better with the input conditions? [Output 1 ⌄]
Question 2: Considering all aspects (semantic relevance, image quality, coherence), which output image do you prefer? [Output 1 ⌄]

[Submit]

Figure 14: Screenshot for human evaluation tasks on the Amazon Mechanical Turk crowdsource evaluation platform.

Meng et al., 2023) propose distillation strategies for diffusion models but they aim at improving inference speed. Our work instead aims to distill the intricate knowledge of teacher models into the student counterparts, ensuring both the improvements over training efficiency and quality retention.

# E    Human Evaluation Interface

We give the human evaluation interface in Figure 14. The human evaluators are mainly asked to finish two tasks and choose their preference from three perspectives.

