## C  Additional Results

### C.1  Additional Qualitative Results on Video Generation

FlexEControl can be further extended to accommodate video generation. In training the controllable video generation model with multiple input conditions, a straightforward strategy is employed to mask out conditions during the training process. In each iteration, a random sample, denoted as $N_s$, is drawn from $[1, N]$ to

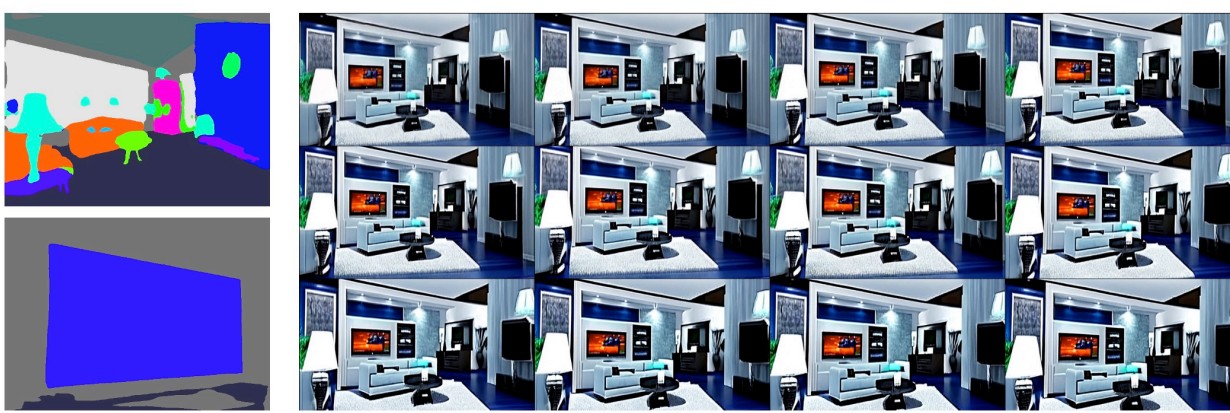

Text Prompt: A TV in the living room

Figure 3: FlexEControl on using multiple conditions for video generation.

determine the number of frames that will incorporate the conditions. Subsequently, $N_s$ unique values are drawn from the set $1, 2, ..., N$, and the conditions are retained for the corresponding frames.

In this section, we showcase the extensibility of FlexEControl in controllable video generation. The results are presented in Figure 2 and Figure 3, where results for providing one condition and multiple conditions are demonstrated.

# D    Training a Small U-Net Backbone

In this section, we discuss further methods to refine the training of a lightweight Stable Diffusion backbone within FlexEControl, aiming to further curtail the number of trainable parameters and minimize memory usage end-to-end. The resulting pre-trained Stable Diffusion backbone, which we denote as FlexEControl-pretraining, offers a more lightweight alternative to the original model while retaining versatility for application in a variety of tasks.

Building upon the strategies delineated in the main paper, we architect a streamlined U-Net structure utilizing low-rank decomposition. This design is complemented by the implementation of knowledge distillation techniques throughout the training process to cultivate an efficient text-to-image generative model. Our training regimen unfolds in two distinct phases: Initially, we focus on establishing a lightweight T2I diffusion model founded on a conventional U-Net framework, with knowledge distillation enhancing this foundational stage. Subsequently, we move to fine-tuning introduced in the main paper, enabling the model to adeptly manage controlled T2I generation tasks. This bifurcated approach yields significant resource savings both in fine-tuning and in the overall model parameter count, setting a new benchmark for efficiency in generative modeling.

## D.1    Background on Low-rank Training

**Background on Training in Low-dimensional Space**    Let $\theta^D = \left[\theta_0{}^D \ldots \theta_m{}^D\right]$ be a set of $m$ $D$-dimensional parameters that parameterize the U-Net within the Stable Diffusion. Instead of optimizing the noise prediction loss in the original parameter space $\left(\theta^D\right)$, we are motivated to train the model in the lower-dimensional space $\left(\theta^d\right)$ (Aghajanyan et al., 2020). Our overall pipeline is trying to train the controllable text-to-image diffusion model in such a lower-dimension space to improve the overall efficiency.

An overview of our proposed two-stage pipeline is shown in Figure 6. We first train the U-Net of a text-to-image model with a low-rank schema. Specifically, we employ matrix factorization techniques that decompose high-dimensional matrices into smaller matrices, capturing essential features with reduced computational overhead. This process is augmented through knowledge distillation, visually represented in green on Figure 6.

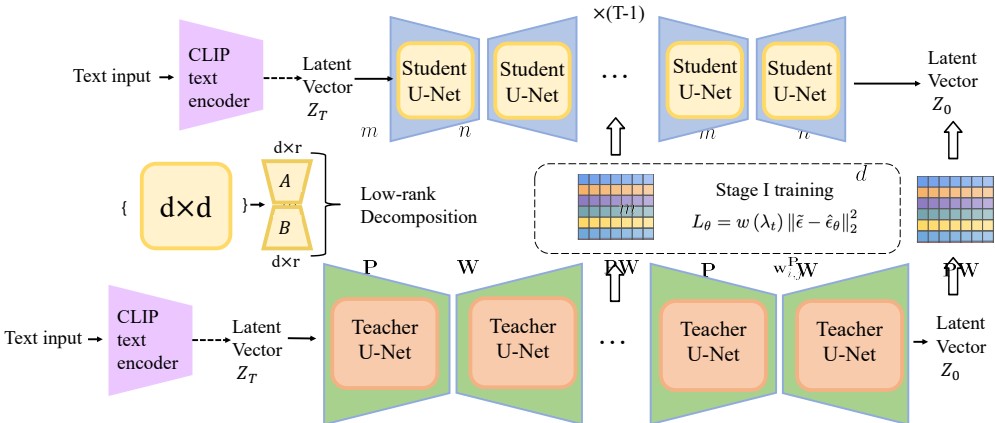

Figure 4: Overview of the Stage-1 training: Training a low-rank U-Net using knowledge distillation from a teacher model (green) to the student model (blue). This process involves initializing the student U-Net with a decomposition into low-rank matrices and minimizing the loss between the predicted noise representations from the student and teacher.

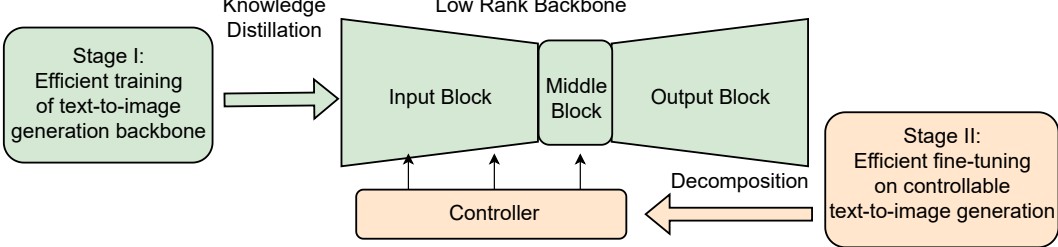

Figure 5: The overview pipeline of our method. Our method improves the efficiency of controllable text-to-image generation from two aspects. At pretraining stage, we propose an efficient pretraining method for the standard text-to-image generation via knowledge distillation. For the finetuning stage introduced in the main paper, we propose to resort to low-rank and Kronecker decomposition to reduce the tunable parameter space.

We then conduct efficient fine-tuning using the methods (shown in the yellow part on Figure 6) with the methods introduced in the main paper, where we employ low-rank decomposition and Kronecker decomposition to streamline the parameter space.

**Low-rank Text-to-image Diffusion Model** To establish a foundational understanding of our model, it's crucial to first comprehend the role of U-Nets in the diffusion process. In diffusion models, there exists an input language prompt $y$ that is processed by a encoder $\tau_\theta$. This encoder projects $y$ to an intermediate representation $\tau_\theta(y) \in \mathbb{R}^{M \times d_\tau}$, where $M$ is denotes the token length, and $d_\tau$ denotes the dimension of the embedding space . This representation is subsequently mapped to the intermediate layers of the U-Net through a cross-attention layer given by

$$\text{Attention}(Q, K, V) = \text{softmax}\left(\frac{QK^T}{\sqrt{d}}\right)V, \tag{2}$$

with $Q = \mathbf{W}_Q \varphi_i(z_t)$, $K = \mathbf{W}_K \tau_\theta(y)$, $V = \mathbf{W}_V \tau_\theta(y)$. In this context, $\varphi_i(z_t) \in \mathbb{R}^{N \times d_\epsilon}$ is an intermediate representation of the U-Net. The terms $\mathbf{W}_V \in \mathbb{R}^{d \times d_\epsilon}, \mathbf{W}_Q \in \mathbb{R}^{d \times d_\tau}, \mathbf{W}_K \in \mathbb{R}^{d \times d_\tau}$ represent learnable projection matrices.

Shifting focus to the diffusion process, during the t-timestep, we can represent:

$$K = \mathbf{W}_K \tau_\theta(y) = AB\tau_\theta(y), \tag{3}$$
$$V = \mathbf{W}_V \tau_\theta(y) = AB\tau_\theta(y), \tag{4}$$

where $A$ and $B$ are decomposed low-rank matrices from the cross-attnetion matrices, $d_\tau$ and $d_\epsilon$ denote the dimension for the text encoder and noise space respectively. Conventionally, the diffusion model is trained via minimizing $\mathcal{L}_\theta = \left\|\epsilon - \epsilon_\theta\right\|_2^2$, where $\epsilon$ is the groundtruth noise and $\epsilon_\theta$ is the predicted noise from the model.

Central to our strategy is a knowledge distillation process. This involves guiding a novice or 'Student' diffusion model using feature maps that draw upon the wisdom of a more seasoned 'Teacher' model. A pivotal insight from our study lies in the mathematical congruence between the low-rank training processes across both training phases, unveiling the symmetries in low-rank training trajectories across both phases.

To fully exploit the prior knowledge from the pretrained teacher model while exploiting less data and training a lightweight diffusion model, we propose a new two-stage training schema. The first one is the initialization strategy to inherit the knowledge from the teacher model. Another is the knowledge distillation strategy. The overall pipeline is shown in Figure 4.

### D.2    Initialization

Directly initializing the student U-Net is not feasible due to the inconsistent matrix dimension across the Student and teacher U-Net. We circumvent this by decomposing U-Net into two low-rank matrices, namely $A$ and $B$ for the reconstruction. We adopt an additional transformation to adapt the teacher's U-Net weights to the Student, which leverages the Singular Value Decomposition (SVD) built upon the teacher U-Net. The initialization process can be expressed as:

1. Compute the SVD of the teacher U-Net: Starting with the teacher U-Net parameterized by $\theta_0$, we compute its SVD as $\theta_0 = U\Sigma V^T$.

2. Extract Low-Rank Components: to achieve a low-rank approximation, we extract the first $k$ columns of $U$, the first $k$ rows and columns of $\Sigma$, and the first $k$ rows of $V^T$. This results in matrices $U_k$, $\Sigma_k$, and $V_k^T$ as follows:

$$U_k = \text{first } k \text{ columns of } U, \tag{5}$$
$$\Sigma_k = \text{first } k \text{ rows \& columns of } \Sigma, \tag{6}$$
$$V_k^T = \text{first } k \text{ rows of } V^T \tag{7}$$

3. We then initialize the student U-Net with $U_k\Sigma_k$ and $V_k^T$ that encapsulate essential information from the teacher U-Net but in a lower-rank format.

We observe in practice that such initialization effectively retains the prior knowledge inherited from Teacher U-Net while enabling the student U-Net to be represented in a compact form thus computationally more efficient for later training.

### D.3    Loss Function

We propose to train our Student U-Net with knowledge distillation (Meng et al., 2023) to mimic the behavior of a teacher U-Net. This involves minimizing the loss between the student's predicted noise representations and those of the teacher. To be specific, our training objective can be expressed as:

$$\mathcal{L}_\theta = w(\lambda_t)\left\|\tilde{\epsilon} - \hat{\epsilon}_\theta\right\|_2^2, \tag{8}$$

where $\tilde{\epsilon}$ denotes the predicted noise in the latent space of Stable Diffusion from the teacher model, $\hat{\epsilon}_\theta$ is the corresponding predicted noise from the student model, parameterized by $\theta$, and $w(\lambda_t)$ is a weighting

Table 1: Comparing U-Net models: Original, decomposed, with and without Knowledge Distillation. FlexEControl-Pretraining showcases a promising balance between performance and efficiency. Note that compared with Stable Diffusion, FlexEControl-Pretraining is only trained on 5 million data. FlexEControl-Pretraining beats Decomposed U-Net w/o Distillation interms of FID and CLIP Score, suggesting the effectiveness of our distillation strategy in training the decomposed U-Net.

| Methods | FID↓ | CLIP Score↑ | # Parameters ↓ |
|---|---|---|---|
| Stable Diffusion | 27.7 | 0.824 | 1290M |
| Standard U-Net w/o Distill. | 66.7 | 0.670 | 1290M |
| Decomposed U-Net w/o Distill. | 84.3 | 0.610 | 790M |
| FlexEControl-Pretraining | 45.0 | 0.768 | 790M |

Table 2: Performance and resource metrics comparison of FlexEControl with the baseline Uni-ControlNet. The FlexEControl approach with distillation shows a significant reduction in resource consumption while providing competitive image quality and outperforming in controllability metrics, especially in segmentation maps. The Δ column shows the improvement of FlexEControl (w/o distillation) compared with no distillation.

| | Metrics | Uni-ControlNet | FlexEControl w/o Distill. | FlexEControl w/ Distill. | Δ |
|---|---|---|---|---|---|
| Efficiency | Memory Cost ↓ | 20GB | **11GB** | **11GB** | **0** |
| | # Params. ↓ | 1271M | **536M** | **536M** | **0** |
| Image Quality | FID ↓ | 27.7 | 84.0 | 43.7 | **- 40.3** |
| | CLIP Score ↑ | 0.82 | 0.61 | 0.77 | **+0.16** |
| Controllability | Sketch Maps (CLIP Score)↑ | 0.49 | 0.40 | 0.46 | **+0.06** |
| | Edge Maps (NMSE ) ↓ | 0.60 | 0.54 | 0.57 | **+0.03** |
| | Segmentation Maps (IoU) ↑ | 0.70 | 0.40 | 0.74 | **+0.34** |

function that may vary with the time step $t$. Such an objective encourages the model to minimize the squared Euclidean distance between the teacher and Student's predictions thus providing informative guidance to the Student. We also tried combining the loss with the text-to-image Diffusion loss but using our training objective works better.

### D.4 Experimental Settings

In the pretraining stage, we used the standard training scheme of Stable Diffusion (Rombach et al., 2022) with the classifier-free guidance (Ho and Salimans, 2022). We employed the Stable Diffusion 2.1 [1] model in conjunction with xFormers (Lefaudeux et al., 2022) and FlashAttention (Dao et al., 2022) using the implementation available in HuggingFace Diffusers [2].

### D.5 Results

Table 1 illustrates the comparison between different variations of our method in the pretraining stage, including original U-Net, decomposed low-rank U-Net, and their respective performance with and without knowledge distillation. It is observed that the decomposed low-rank U-Net models demonstrate efficiency gains, with a reduction in the total number of parameters to 790M, although at the cost of some fidelity in metrics such as FID and CLIP Score. Employing distillation helps to mitigate some of these performance reductions.

---

[1] https://huggingface.co/stabilityai/stable-diffusion-2-1
[2] https://huggingface.co/docs/diffusers/index

You will see **two input images (edge maps) and a text prompt,** along with **two generated output images**.
The goal is to generate images that align with both the input images and the text prompt.

Your task is to evaluate the two generated images based on the following criteria:
[1] Alignment with Input: Which output image aligns better with the input conditions (edge maps)?
[2] Overall Preference: Considering all aspects, which output image do you prefer? This includes:
    a) Semantic relevance: Does the output image align well with the text prompt?
    b) Image quality: Is the output image of good visual quality?
    c) Coherence: Does the output image properly reflect the edges shown in the input images?

---

Question 1: Which output image aligns better with the input conditions? `Output 1 ∨`
Question 2: Considering all aspects (semantic relevance, image quality, coherence), which output image do you prefer? `Output 1 ∨`

`Submit`

Figure 6: Screenshot for human evaluation tasks on the Amazon Mechanical Turk crowdsource evaluation platform.

Table 2 illustrates the comparison between FlexEControl including pretraining and the baseline training end-to-end. It is observed that the decomposed low-rank U-Net models demonstrate efficiency gains, with a reduction in the total number of parameters to 536M, although at the cost of some fidelity in metrics such as FID and CLIP Score. Employing distillation helps to mitigate some of these performance reductions.

These collective results affirm our method's capability to not only enhance efficiency but also improve or maintain performance across various aspects of text-to-image generation.

## E   Additional Related Works

**Knowledge Distillation for Vision-and-Language Models**   Knowledge distillation (Gou et al., 2021), as detailed in prior research, offers a promising approach for enhancing the performance of a more streamlined "student" model by transferring knowledge from a more complex "teacher" model (Hinton et al., 2015; Sanh et al., 2019; Hu et al.; Gu et al., 2021; Li et al., 2021). The crux of this methodology lies in aligning the predictions of the student model with those of the teacher model. While a significant portion of existing knowledge distillation techniques leans towards employing pretrained teacher models (Tolstikhin et al., 2021), there has been a growing interest in online distillation methodologies (Wang and Jordan, 2021). In online distillation (Guo et al., 2020), multiple models are trained simultaneously, with their ensemble serving as the teacher. Our approach is reminiscent of online self-distillation, where a temporal and resolution ensemble of the student model operates as the teacher. This concept finds parallels in other domains, having been examined in semi-supervised learning (Peters et al., 2017), label noise learning (Bengio et al., 2010), and quite recently in contrastive learning (Chen et al., 2020). Our work on distillation for pretrained text-to-image generative diffusion models distinguishes our method from these preceding works. (Salimans and Ho, 2022; Meng et al., 2023) propose distillation strategies for diffusion models but they aim at improving inference speed. Our work instead aims to distill the intricate knowledge of teacher models into the student counterparts, ensuring both the improvements over training efficiency and quality retention.

## F   Human Evaluation Interface

We give the human evaluation interface in Figure 6. The human evaluators are mainly asked to finish two tasks and choose their preference from three perspectives.