# OpenReview forum: "FlexEControl: Flexible and Efficient Multimodal Control for Text-to-Image Generation"
_TMLR — Accepted by TMLR_

### Review · Reviewer_moeq · 2024-06-29

**Summary Of Contributions:**

One of the main applications of diffusion based text-to-image are ControlNets, i.e., the possibility to provide additional input to control the generated image. Usually an additional network, called ControlNet is trained and the additional input controls the structure of the generated image.
According to the paper, current ControlNets struggle when multiple inputs are involved. In addition, the paper claims to improve efficiency.
The proposed framework, FlexEControl, is built upon a unique weight decomposition strategy. The approachachieves a reduction of 41% in trainable parameters and 30% in memory usage compared to a similar method - UniControl.
The paper also suggests augmenting the training data for multiple conditions by segmenting different objects separately and then using a novel loss to make sure the network handles the different segments.

**Audience:**

Yes

**Broader Impact Concerns:**

There are no ethical concerns

**Claims And Evidence:**

Yes

**Requested Changes:**

Overall, the paper is not well-written in my opinion.
All weaknesses above (1-7) should be addressed to improve the paper.

**Strengths And Weaknesses:**

Weaknesses:

1. "copied Stable Diffusion encoder" - I consider myself an expert in diffusion models but not sure what the meaning of stable diffusion encoder is? VAE? Part of the Unet? ControlNet?

2. Figure 3 - it is better if the figure caption is clear even without diving into the text. What is the purpose of this figure? It would be better to add this to the figure itself

3. After reading the Method section and the related work section I'm not sure how UniControl works and what is the difference from UniControl? Since UniControl seems like the most related work I find it important

4. The ablation study is not convincing. I want to understand what happens when lambdas are zero. Please convince me that the complicated losses are useful.

5. Figure 6 - the effect of the segmentation seems kind of weak. Although other methods fail as well - this is kind of disappointing. I would consider this example as a failure case.

6. Actually, I think the limitation/failure cases section would improve the paper significantly.

7. Conclusions section is too short and lacks novel insights.


Strengths:

1. The task of multi-control generation is interesting and highly applicable
2. The losses and the specific decomposition are novel and interesting
3. The evaluation seems adequate, the model is compared to several baselines and several metrics are evaluated.

---

> ### Author Response · Authors · 2024-08-22
> **Author Response to Reviewer moeq (1/2)**
>
> Thank you for your comprehensive review and for bringing up weaknesses that need clarification or improvement. We have now carefully addressed each of your points in our revision.
>
>
>
>
> >- "copied Stable Diffusion encoder" - I consider myself an expert in diffusion models but not sure what the meaning of stable diffusion encoder is? VAE? Part of the Unet? ControlNet?
>
>
> We apologize for the confusion. The term "copied Stable Diffusion encoder" refers to the trainable copy of the Stable Diffusion encoder block and the Stable Diffusion middle block (the down projection structure and the middle block in the U-Net of Stable Diffusion). We borrowed this terminology from [1], where it was introduced. We have now clarified this in the revision.
>
>
> [1] Adding Conditional Control to Text-to-Image Diffusion Models
>
>
>
>
> >- Figure 3 - it is better if the figure caption is clear even without diving into the text. What is the purpose of this figure? It would be better to add this to the figure itself
>
>
> In Figure 3, we intend to demonstrate that the Kronecker Decomposition possesses the multiplicative rank property, distinguishing it from techniques such as LoRA. Additionally, we illustrate the content-preserving qualities of the Kronecker Product. For example, the shared “slow” weights of an image, when combined with another set of “fast” low-rank weights, can preserve the original image’s distribution without losing semantic integrity, as shown in Figure 3. We have revised the figures to be more self-contained and added more examples to the revision to make the purpose of the figure clearer.
>
>
>
>
>
>
>
>
>
>
> >- After reading the Method section and the related work section I'm not sure how UniControl works and what is the difference from UniControl? Since UniControl seems like the most related work I find it important
>
>
> UniControl employs an MOE-Style Adapter, consisting of a group of convolution modules to capture features of various low-level visual conditions. In contrast, our approach focuses on handling multimodal control, particularly in scenarios with multiple multimodal conditions, either from the same type (e.g., two segmentation maps + text conditioning) or different types (e.g., segmentation maps + edge maps + text conditioning), especially when these conditions conflict. While UniControl also uses controls from differen conditions, our method differs significantly in its approach.
>
> Methodologically, we propose a Kronecker Decomposition strategy that amalgamates diverse conditions with a common set of weights, improving efficiency. Additionally, we introduce a Dataset Augmentation with Text Parsing and Segmentation strategy, along with Cross-Attention Supervision and Masked Diffusion Loss during training, to enhance the controllability of our framework. These strategies collectively form our method for multimodal controllable text-to-image generation.
> We also want to clarify that our work presents a general approach that can be applied to UniControl or other backbones. We have added this discussion to the related work section in our revision.

---

> ### Author Response · Authors · 2024-08-22
> **Author Response to Reviewer moeq (2/2)**
>
> >- The ablation study is not convincing. I want to understand what happens when lambdas are zero. Please convince me that the complicated losses are useful.
>
>
> We want to clarify that when the lambdas are set to zero, it removes the Cross-Attention Supervision Loss and Masked Diffusion Loss. We evaluated these cases in Table 3, and as shown, both perform worse than FlexEControl  with these losses included. To further clarify, we have added qualitative results (Figure 4) demonstrating the effects when the lambdas are zero (i.e., when the Cross-Attention Supervision and Masked Diffusion Losses are not used). These results, along with the additional explanations, have been included in the revision.
>
>
>
>
>
>
>
>
>
>
>
>
> >- Figure 6 - the effect of the segmentation seems kind of weak. Although other methods fail as well - this is kind of disappointing. I would consider this example as a failure case.
> >- Actually, I think the limitation/failure cases section would improve the paper significantly.
>
> We have added additional qualitative results in Figures 6, 8, and 9 of the revision to further illustrate the effect of segmentation. Additionally, we have included a dedicated limitation/failure cases section to discuss the weaknesses of FlexEControl and provide explanations for these issues.
>
>
>
>
>
>
>
> >- Conclusions section is too short and lacks novel insights.
>
>
> We have expanded and rewritten the conclusion section to address this concern. Specifically, we have detailed how FlexEControl addresses the challenges of controllable text-to-image generation conditioned on multimodal inputs. Our approach introduces several key innovations, including Dataset Augmentation with Text Parsing and Segmentation, Cross-Attention Supervision, and Masked Diffusion Loss, which collectively enhance the flexibility of text-to-image generation under various conditions. Additionally, we propose an Efficient Training strategy that improves parameter, data, and memory efficiency without compromising performance. The revised conclusion also emphasizes the potential for extending our method to other architectures and suggests exploring more advanced efficient training techniques in future work to further optimize model efficiency and performance.
>
>
>
>
> We hope our responses have addressed your concerns and provided clarity on the points you raised. If you have any further questions or need additional clarification, we are happy to provide further details. Thank you again for your valuable feedback!

---

### Review · Reviewer_3622 · 2024-07-22

**Summary Of Contributions:**

This work aims to merge parameter efficiency and multiple controls in image synthesis using Text-to-Image LDMs. The authors claim high parameter efficiency with of decrease of 41% parameters compared to the SOTA, Uni-Controlnet. Broadly, the authors use Kronecker Decomposition

The authors introduce 2 new loss functions for better controllable generation and semantic understanding.

Overall, this work is a good mix of multi-control generation equipped with parameter efficiency and exhibits good results both qualitatively and quantitatively.

**Audience:**

Yes

**Broader Impact Concerns:**

No concerns on broader impact.

**Claims And Evidence:**

Yes

**Requested Changes:**

This work has good potential but I am not completely satisfied with the presentation. I tried to provide constructive feedback in the section above and here are some suggestions that would make the paper better IMO.

1. Proper citations, at least to [2] and [3] must be added and should be mentioned in the related works.

2. In __2.2__, instead of citing previous works repeatedly, the authors are requested to mention the multiplicative rank property along with properties showing the content-preserving qualities of the Kronecker Product.

3. Instead of wasting much space in explaining terms commonly known to most researchers(__3.2__, __3.3__), the authors must bring implementation details of LoRA, PHM, etc. from the appendix to the main paper to bring more clarity.

4. Qualitative results on lambda values of loss functions could better justify the addition of new terms and the selection of lambda values for training.

5. Unlike other model names, the authors forgot to emphasize __improved_aesthetics_6plus__ in __3.1__ and __3.6.1__. It is good to keep it consistent with other model names.😊

**Strengths And Weaknesses:**

Strengths:
1. I like the idea of merging parameter-efficient training with multiple controls. In my opinion, this work can have multiple applications such as label-preserving image generation for autonomous driving. The qualitative results show the affinity of generations to the control prompts.

2. The idea of introducing __Masked Diffusion Loss__ looks excellent to me. This can surely improve the spatial understanding of diffusion models allowing them to better understand the text prompts and hence generate better images.

3. This work explores Kronecker Decomposition which is an underexplored matrix decomposition technique having excellent content preservation qualities.💯

Weaknesses:
This work shows interesting results and findings however I have a few concerns mentioned below.

1. This work seems to be highly motivated by FreeControl[1]. Although FreeControl is a training-free approach, the ideologies seem to be quite similar, but I find no mention or comparison to [1] in the paper. Moreover, the authors claim novelty in applying Kronecker Decomposition for parameter efficiency but do not cite works which have already explored Kronecker Product for tuning T2I models[2][3].

2. In section __2.2__, the authors cite previous works but do not show any concrete reason of using Kronecker Products for matrix decomposition. There is no mention of __multiplicative rank__ property of Kronecker Products which sets them apart from techniques such as LoRA.🤦

3. In section __2.3__, paragraph __Cross-Attention Supervision__, the authors do not clearly mention the benefit of adding the extra loss term. In my opinion, __Masked Diffusion Loss__ already covers the semantic-spatial understanding of concepts present in the image.

4. The authors do not provide any information about the implementation of baselines used for comparison in the main paper. I can only find mention of LoRA (Hu et al.), PHM(Zhang et al.), etc, in the main paper. Instead, they waste a lot of space explaining __Evaluation Metrics__ and __Input Conditions__. I am strongly against comparing architectures using tables and figures without mentioning the implementation. It is good that the authors have mentioned a brief section about it in the appendix.

5. In __Figure 5__, the authors compare their method with ControlNet, however, no quantitative evaluation is provided with ControlNet.

6. Since the authors introduce 2 new loss terms, a qualitative comparison showing the effect of each term was expected which the authors have not provided. They only included __Figure 4__ as an ablation on lambda values.


[1]: FreeControl: Training-Free Spatial Control of Any Text-to-Image Diffusion Model with Any Condition
[2]: DiffuseKronA: A Parameter Efficient Fine-tuning Method for Personalized Diffusion Model
[3]: Navigating Text-To-Image Customization:From LyCORIS Fine-Tuning to Model Evaluation

---

> ### Author Response · Authors · 2024-08-22
> **Author Response to Reviewer 3622 (1/2)**
>
> We really appreciate your detailed feedback and constructive comments, which have helped us improve our paper. Below are our responses to your points raised:
>
> >- This work seems to be highly motivated by FreeControl[1]. Although FreeControl is a training-free approach, the ideologies seem to be quite similar, but I find no mention or comparison to [1] in the paper. Moreover, the authors claim novelty in applying Kronecker Decomposition for parameter efficiency but do not cite works which have already explored Kronecker Product for tuning T2I models[2][3].
> >- Proper citations, at least to [2] and [3] must be added and should be mentioned in the related works.
>
>
>
>
>
>
>
>
> We want to clarify that our work is distinct from FreeControl in several key aspects. First, our primary focus is on enhancing the flexibility of multimodal control, particularly in handling multiple conditions, whether from the same modality (e.g., two segmentation maps + text conditioning) or different modalities (e.g., segmentation maps + edge maps + text conditioning). This flexibility is crucial, especially when these conditions conflict, and we have added more examples in the revision to demonstrate this. Technically, we propose an Enhanced Training strategy for Conditional Input, which combines Dataset Augmentation with Text Parsing and Segmentation, Cross-Attention Supervision, and Masked Diffusion Loss.
>
>
>
>
> Second, unlike FreeControl, which is a training-free approach, our method is training-based and achieves **significantly better performance**. We have added a performance comparison between FreeControl and FlexEControl, as shown below:
>
>
> | Models                       | Canny (SSIM) ↑ | MLSD (SSIM) ↑ | HED (SSIM) ↑ | Sketch (SSIM) ↑ | Depth (MSE) ↓ | Segmentation (mIoU) ↑ | Poses (mAP) ↑ | FID ↓ | CLIP Score ↑ |
> |------------------------------|----------------|---------------|--------------|-----------------|--------------|------------------------|--------------|-------|--------------|
> | FlexEControl (ours)           | **0.4990**     | **0.6385**    | **0.5041**   | **0.5518**      | **90.93**     | **0.7496**             | **0.2093**   | **27.55** | **0.4963**  |
> | FreeControl                   | 0.4283         | 0.5580        | 0.4631       | 0.4688          | 91.92        | 0.5397                 | 0.1428       | 27.99 | 0.4811       |
>
>
>
>
>
>
> Regarding [2] and [3], it’s important to note that [2] primarily focuses on subject-driven text-to-image generation, where the input image conditions involve several images of the same subject. In contrast, FlexEControl is capable of handling different conditions from various modalities (e.g., segmentation maps, sketch maps) and flexibly composing them. While [3] explores the Kronecker Product for tuning T2I models, it also mainly addresses subject-driven and style-preserving text-to-image generation. Our work is the first to apply the Kronecker Product for multimodal control in text-to-image generation, significantly improving both efficiency and flexibility. We have now included these comparisons and citations in the revised manuscript
>
>
> [2] DiffuseKronA: A Parameter Efficient Fine-tuning Method for Personalized Diffusion Model
>
> [3] Navigating Text-To-Image Customization:From LyCORIS Fine-Tuning to Model Evaluation
>
>
>
>
>
>
> >- In section 2.2, the authors cite previous works but do not show any concrete reason of using Kronecker Products for matrix decomposition. There is no mention of multiplicative rank property of Kronecker Products which sets them apart from techniques such as LoRA.🤦
>
>
> >- In 2.2, instead of citing previous works repeatedly, the authors are requested to mention the multiplicative rank property along with properties showing the content-preserving qualities of the Kronecker Product.
>
>
> Thank you for the suggestion! We have revised this section to address your concerns. We use Kronecker Products for matrix decomposition because they not only reduce parameter and memory costs, as well as computation, as shown in Table 1, but also possess the multiplicative rank property. This property sets Kronecker Products apart from techniques like LoRA, enhancing data efficiency in multimodal controllable T2I by allowing different conditions to reuse the same data. Additionally, the multiplicative rank property is crucial for maintaining content-preserving qualities during decomposition. We have added further details and examples in Figure 3 to illustrate these properties, and these updates have been included in the revision.

---

> ### Author Response · Authors · 2024-08-22
> **Author Response to Reviewer 3622 (2/2)**
>
> >- In section 2.3, paragraph Cross-Attention Supervision, the authors do not clearly mention the benefit of adding the extra loss term. In my opinion, Masked Diffusion Loss already covers the semantic-spatial understanding of concepts present in the image.
>
>
>
>
> We introduced cross-attention supervision to strengthen the alignment between text prompts and image feature maps during the diffusion process. While the Masked Diffusion Loss focuses on fitting the estimated noise to the ground truth noise in the specified region, the cross-attention supervision explicitly enhances the guidance of text throughout the entire diffusion process via cross-attention maps. We have clarified this in the revision. Additionally, in Table 3, we show that the model without Cross-Attention Supervision or Masked Diffusion Loss performs worse than FlexEControl with both, demonstrating the effectiveness of these loss terms. We have also now added qualitative comparisons in Figure 4 (b) to show the impact of each term in the revision.
>
>
>
>
>
>
> >- The authors do not provide any information about the implementation of baselines used for comparison in the main paper. I can only find mention of LoRA (Hu et al.), PHM(Zhang et al.), etc, in the main paper. Instead, they waste a lot of space explaining Evaluation Metrics and Input Conditions. I am strongly against comparing architectures using tables and figures without mentioning the implementation. It is good that the authors have mentioned a brief section about it in the appendix.
> >- Instead of wasting much space in explaining terms commonly known to most researchers(3.2, 3.3), the authors must bring implementation details of LoRA, PHM, etc. from the appendix to the main paper to bring more clarity.
>
>
>
>
> Thank you for this suggestion! We have restructured sections 3.2 and 3.3 to include the implementation details of LoRA, PHM, and other baselines directly in the main paper, bringing more clarity to the comparisons. We have also moved the redundant introduction of input conditions and evaluation metrics to the appendix to save space.
>
>
>
>
>
>
>
>
> >- In Figure 5, the authors compare their method with ControlNet, however, no quantitative evaluation is provided with ControlNet.
>
>
> We have now added quantitative evaluations comparing our method with ControlNet in Table 2, where we provide a quantitative assessment for single structural conditional inputs. The results are as follows:
> | Models            | Canny (SSIM) ↑ | MLSD (SSIM) ↑ | HED (SSIM) ↑ | Sketch (SSIM) ↑ | Depth (MSE) ↓ | Segmentation (mIoU) ↑ | Poses (mAP) ↑ | FID ↓  | CLIP Score ↑ |
> |-------------------|----------------|---------------|--------------|-----------------|--------------|------------------------|--------------|--------|--------------|
> | FlexEControl (ours)| **0.4990**     | **0.6385**    | **0.5041**   | 0.5518          | 90.93        | **0.7496**             | **0.2093**   | **27.55** | **0.4963**  |
> | ControlNet         | 0.4989         | 0.6172        | 0.4990       | **0.6013**      | **89.08**    | 0.7481                 | 0.2024       | 27.62  | 0.4931       |
>
> As shown, FlexEControl outperforms ControlNet across most evaluation metrics, despite ControlNet being specifically trained for single conditional input.
>
>
>
>
>
>
> >- Since the authors introduce 2 new loss terms, a qualitative comparison showing the effect of each term was expected which the authors have not provided. They only included Figure 4 as an ablation on lambda values.
>
> >- Qualitative results on lambda values of loss functions could better justify the addition of new terms and the selection of lambda values for training.
>
> We have now added a qualitative comparison showing the effect of the two loss terms in Figure 4 (b) of the revision, which better justifies the addition of these new terms and the importance of lambda values for training.
>
>
>
>
>
>
>
>
>
>
>
>
> >- Unlike other model names, the authors forgot to emphasize improved_aesthetics_6plus in 3.1 and 3.6.1. It is good to keep it consistent with other model names.😊
>
> Thanks for the reminder! We have now emphasized "improved_aesthetics_6plus" and ensure its consistency with other model names in the revision. We want to assure you that all models are kept consistent and in the same setting.
>
>
>
>
>
>
>
>
> We hope our responses address your concerns and provide a clearer understanding of our work. If you have any further questions or require additional clarification, we are happy to provide further details.

---

### Review · Reviewer_UsVX · 2024-08-08

**Summary Of Contributions:**

This paper targets for flexible and efficient controllable text-to-image generation. To achieve that, the paper proposes a weight decomposition method to allow for streamlined integration of various input conditions. Experiments demonstrate that it can reduce computational resources and support various modalities conditions.

**Audience:**

Yes

**Broader Impact Concerns:**

N.A. Already included in the paper.

**Claims And Evidence:**

Yes

**Requested Changes:**

Please refer to the weaknesses and questions.

**Strengths And Weaknesses:**

**Strengths**

1.	The paper introduces the Kronecker product operation to compute the shared decomposed low-rank subspace, which effectively reduces the trainable parameters. Introducing the slow and fast low-rank weights into multi-modality conditional generation is reasonable.

2.	The proposed method achieves trainable parameters reduction and less GPU memory consumption, with comparable performances compared to SOTAs.

3.	The paper is well written and motivated.

**Weaknesses**

1.	Although introducing Kronecker product operation for parameter reduction is reasonable and interesting, it seems to lack technical contributions that are unique to the text-to-image generation. The authors are suggested to include more analysis on the unique contributions or designs that are more tailored for T2I generation.

2.	The dataset augmentation part relies on the CLIPSeg to obtain masks, which could be less accurate when different objects share similar semantics information or are close to each other. This may also result in inferior results when multiple objects are occluded.

**Questions**

1.	For multi-modal conditional T2I generation, the paper mainly showed results conditioned on 2 conditions. Could the method support more than 2 conditions?

2.	Could the authors show some failure cases analysis?

3.	For multi-condition generation, the paper mainly reports foreground and background generation. Is it possible to generate two foregrounds?

---

> ### Author Response · Authors · 2024-08-22
> **Author Response to Reviewer UsVX (1/1)**
>
> Thank you for your constructive feedback and for recognizing the effectiveness, efficiency, and motivation of our method! We appreciate the opportunity to clarify and extend the explanations of our methods and results.
>
>
>
> > - Although introducing Kronecker product operation for parameter reduction is reasonable and interesting, it seems to lack technical contributions that are unique to the text-to-image generation. The authors are suggested to include more analysis on the unique contributions or designs that are more tailored for T2I generation.
>
>
>
> We would like to clarify that the Kronecker Decomposition proposed in this work is specifically designed and tailored for text-to-image generation. Our approach differs from its application in other tasks because the decomposed weights are shared across different controlled conditions, introducing a novel method for handling multimodal inputs. This approach is unique to our method and brings new insights into enhancing controllability in text-to-image generation. To clarify this further, we have redrawn Figure 2 in the revision to better illustrate the Kronecker Decomposition process.
>
> Additionally, we would like to highlight the technical contributions that are unique to our method for improving controllability under multiple flexible conditions:
>
> - Dataset Augmentation with Text Parsing and Segmentation: We utilize GPT-3.5-turbo to filter and parse text prompts that contain multiple object entities. For these processed data, we apply CLIPSeg to obtain corresponding segmentation maps from the images, thereby augmenting the dataset.
> - Cross-Attention Supervision: During training, we incorporate cross-attention supervision alongside the noise prediction loss to ensure focused reconstruction in areas relevant to each text-derived concept.
> - Masked Diffusion Loss: This loss improves fidelity to the specified conditions during FlexEControl's training. Compared to the original diffusion training loss, it helps the model concentrate on the local conditional regions.
>
>
> >- The dataset augmentation part relies on the CLIPSeg to obtain masks, which could be less accurate when different objects share similar semantics information or are close to each other. This may also result in inferior results when multiple objects are occluded.
>
> Our motivation for proposing dataset augmentation, along with cross-attention supervision, is to enhance the controllability of text-to-image generation (T2I) in scenarios with multiple conditions, particularly when these conditions share similar semantics that may conflict. The segmentation masks generated during the dataset augmentation process are selectively used to improve training performance, specifically when there is a clear, single mask corresponding to a given object name. This approach ensures that the dataset remains robust and focused. To further validate this, we analyzed the augmented dataset using GPT-4 to identify objects with similar semantics and found that only 0.7% of the sentences contain such objects, indicating a minimal impact on accuracy. We have clarified these points in the revision.
>
>
>
>
> > - For multi-modal conditional T2I generation, the paper mainly showed results conditioned on 2 conditions. Could the method support more than 2 conditions?
>
> While our paper primarily shows results conditioned on two inputs as a general scenario, we emphasize that our method is versatile and can support **multiple conditions (more than 2)**. We have now added results for scenarios involving more than two conditions in the revision, as shown in Figure 9 of the revision. Here, we demonstrate the successful application of our method to three segmentation maps and canny edge maps as conditional inputs, confirming its capability to handle multiple conditions effectively.
>
>
>
> > - Could the authors show some failure cases analysis?
>
> We have included a failure case analysis (Figure 10) in the revision. Our method performs relatively worse in human generation, primarily due to the limited availability of human portrait images in the training dataset and the limitations of the Stable Diffusion backbone used in FlexEControl in handling such cases. However, as a general approach, FlexEControl could benefit from a stronger pretrained model.
>
>
>
>
>
>
> > - For multi-condition generation, the paper mainly reports foreground and background generation. Is it possible to generate two foregrounds?
>
> We have now added qualitative results showing the generation of two foregrounds (Figure 8) in the revision. Additionally, we have demonstrated cases with three foregrounds in Figure 9. These results demonstrate that FlexEControl effectively handles and generates multiple foregrounds.
>
>
>
>
>
>
>
> We hope our responses address your concerns and provide a clearer understanding of our contributions. If you have any further questions or require additional information, we would be happy to provide further clarification!

---

### Author Response · Authors · 2024-08-22
**General Response to All Reviewers**

Dear Reviewers,

We sincerely thank you for your thorough and insightful reviews of our paper! Your valuable feedback has greatly helped in improving the quality and clarity of our work. We have carefully considered each of your comments and made substantial revisions to address your concerns.

We are also pleased to inform you that we have uploaded a new version of the manuscript, with all changes and improvements highlighted in blue for easy identification. We have also provided a point-by-point response to each of your comments, addressing the specific issues raised and explaining how we have revised the paper accordingly.

Please feel free to contact us if you have any further questions or need additional clarification.

Sincerely,

Authors of Paper 2559

---

### Author Response · Authors · 2024-10-29
**Follow-up on the Review Progress for TMLR Paper 2559**

Dear Action Editors,

We hope this message finds you well. We are writing to follow up regarding the review progress of our submission, TMLR Paper 2559. It has been over six months since our submission and two months since our rebuttal, and we have not yet received any further feedback or decision on our submission. We understand that the review process can take time, but we would greatly appreciate any updates on the current status of our paper.

If there are any additional steps we can take on our end to help expedite the process, we would be grateful to know.

Thank you once again for your time and consideration. We look forward to hearing from you!

Best regards,

Authors

---

### Decision · Action_Editor_Dp2J · 2024-11-15

**Recommendation:** Accept as is

**Comment:**

This work aims to merge parameter efficiency and multiple controls in image synthesis using Text-to-Image LDMs. The authors claim high parameter efficiency with of decrease of 41% parameters compared to the SOTA, Uni-Controlnet. Broadly, the authors use Kronecker Decomposition

The authors introduce two loss functions for better controllable generation and semantic understanding.

All three reviewers are positive about the paper and are satisfied with the rebuttal. This work is a good mix of multi-control generation equipped with parameter efficiency and exhibits good results both qualitatively and quantitatively.

**Audience:**

Yes, text-to-image generation is an important task.

**Claims And Evidence:**

Yes. The paper provided solid experiments in their manuscript and rebuttal to support their claimed parameter and memory reduction.

---

> ### Author Response · Authors · 2024-12-13
> **Upload of Camera Ready Revision**
>
> We sincerely thank all reviewers and AE for the insightful comments! The camera-ready revision has been uploaded with all the comments addressed.